# LIKELIHOOD ADJUSTED SEMIDEFINITE PROGRAMS FOR CLUSTERING HETEROGENEOUS DATA

## ABSTRACT

Clustering is a widely deployed unsupervised learning tool. Model-based clustering is a flexible framework to tackle data heterogeneity when the clusters have different shapes. Likelihood-based inference for mixture distributions often involves non-convex and high-dimensional objective functions, imposing difficult computational and statistical challenges. The classic expectation-maximization (EM) algorithm is a computationally thrifty iterative method that maximizes a surrogate function minorizing the log-likelihood of observed data in each iteration, which however suffers from bad local maxima even in the special case of the standard Gaussian mixture model with common isotropic covariance matrices. On the other hand, recent studies reveal that the unique global solution of a semidefinite programming (SDP) relaxed $K$-means achieves the information-theoretically sharp threshold for perfectly recovering the cluster labels under the standard Gaussian mixture model. In this paper, we extend the SDP approach to a general setting by integrating cluster labels as model parameters and propose an iterative likelihood adjusted SDP (iLA-SDP) method that directly maximizes the *exact* observed likelihood in the presence of data heterogeneity. By lifting the cluster assignment to group-specific membership matrices, iLA-SDP avoids centroids estimation – a key feature that allows exact recovery under well-separateness of centroids without being trapped by their adversarial configurations. Thus iLA-SDP is less sensitive than EM to initialization and more stable on high-dimensional data. Our numeric experiments demonstrate that iLA-SDP can achieve lower mis-clustering errors over several widely used clustering methods including $K$-means, SDP and EM algorithms.

## 1 INTRODUCTION

Clustering analysis has been widely studied and regularly used in machine learning and its applications in network science (Girvan & Newman, 2002), computer vision (Shi & Malik, 2000; Joulin et al., 2010), manifold learning (Chen & Yang, 2021a) and bioinformatics (Karim et al., 2020). Perhaps by far the most popular clustering method is the $K$-means (MacQueen, 1967) partially because there are computationally convenient algorithms such as Lloyd's algorithm and $K$-means++ for heuristic approximation (Lloyd, 1982; Arthur & Vassilvitskii, 2007). Mathematically, $K$-means aims to find the optimal partition of data to minimize the total within-cluster squared Euclidean distances, which is equivalent to the maximum profile likelihood estimator under the standard Gaussian mixture model (GMM) with common isotropic covariance matrices (Chen & Yang, 2021b). Nevertheless, real data usually exhibit various degrees of heterogeneous features such as the cluster shapes may vary from component to component, which renders $K$-means as a sub-optimal clustering method.

Another popular clustering method is the classic expectation-maximization (EM) algorithm, which is a computationally thrifty method based on the idea of data augmentation to iteratively optimize the non-convex observed data likelihood (Dempster et al., 1977). Theoretical investigations reveal that the EM algorithm suffers from bad local maxima even in the one-dimensional standard GMM with well-separated cluster centers (Jin et al., 2016). Thus practically even when applied in highly favorable separation-to-noise ratio settings, careful initialization, often through multiple random initializations or a warm-start by another heuristic method such as hierarchical clustering (Fraley & Raftery, 2002), is the key for the EM algorithm to find the correct cluster labels and model parameters. With a reasonable initial start, the EM algorithm has been shown to achieve good statistical properties (Balakrishnan et al., 2017; Wu & Zhou, 2019).

In this paper, we consider the likelihood-based inference to tackle the problem of recovering cluster labels in the presence of data heterogeneity. Our motivation stems from the recent progress in understanding the computational and statistical limits for convex relaxation methods of the $K$-means clustering. Since $K$-means is a worst-case NP-hard problem (Aloise et al., 2009), various heuristic approximation algorithms such as Lloyd's algorithm (Lloyd, 1982; Lu & Zhou, 2016), and computationally tractable relaxations such as spectral clustering (Meila & Shi, 2001; Ng et al., 2001; Vempala & Wang, 2004; Achlioptas & McSherry, 2005; von Luxburg, 2007; von Luxburg et al., 2008) and semidefinite programs (SDP) (Peng & Wei, 2007; Mixon et al., 2016; Li et al., 2017; Fei & Chen, 2018; Chen & Yang, 2021a; Royer, 2017; Giraud & Verzelen, 2018; Bunea et al., 2016; Zhuang et al., 2022a), have been proposed in literature. Among the existing solutions, the SDP approach is particularly attractive in that it attains information-theoretically optimal threshold on centroid separations for exact recovery of cluster labels (Chen & Yang, 2021b).

**Our contributions.** We extend the SDP approach to a general setting with heterogeneous features by integrating cluster labels as model parameters (together with other component-specific parameters) and propose an *iterative likelihood adjusted SDP* (iLA-SDP) method that directly maximizes the *exact* observed data likelihood. Our idea is to tailor the strength of SDP relaxation of the $K$-means clustering method in the isotropic covariance case for likelihood-awareness inference. On one hand, iLA-SDP has a similar flavor as the EM algorithm by maximizing the likelihood function of the observed data. On the other hand, different from the EM framework, iLA-SDP treats the cluster labels as *unknown* parameters while profiles out the cluster centers (i.e., centroids), which brings several statistical and algorithmic advantages.

First in the arguably simplest one-dimensional GMM setting, EM is known to fail in certain configurations of centroids even when they are well-separated (Jin et al., 2016). In other words, EM is sensitive to initialization and model configuration. The main reason is due to the effort for estimating the cluster centers during the EM iterations. In iLA-SDP, cluster centers are regarded as nuisance parameters and profiled out to obtain a likelihood function in component-specific parameters including only the cluster covariance matrices. Thus iLA-SDP is more stable and performs empirically better than EM.

Second, cluster labels in EM are latent variables that are estimated by their posterior probabilities and the observed log-likelihood for component parameters and mixing weights are optimized through minorizing functions during iterations. In iLA-SDP, cluster labels are regarded as parameters optimized through the likelihood function jointly in the labels and covariance matrices. Thus iLA-SDP is a more direct approach than EM for taming the non-convexity in the observed log-likelihood objective and we prove that it perfectly recovers the true clustering structure if the clusters are well-separated under a lower bound without concerning the configurations of centroids.

The rest of the paper is organized as follows. In Section 2, we review some background on partition-based formulation for model-based clustering. In Section 3, we introduce the likelihood adjusted SDP for recovering the true partition structure and discuss its connection to the EM algorithm. In Section 4, we compare the performance of several widely used clustering methods on two real datasets.

## 2 MODEL-BASED CLUSTERING: A PARTITION FORMULATION

We consider the model-based clustering problem. Suppose the data points $X_1, \ldots, X_n \in \mathbb{R}^p$ are independent random variables sampled from $K$-component Gaussian mixture model (GMM). Specifically, let $G_1^*, \ldots, G_K^*$ be the true partition of the index set $[n] := \{1, \ldots, n\}$ such that if $i \in G_k^*$, then

$$X_i = \mu_k + \epsilon_i, \tag{1}$$

where $\mu_k \in \mathbb{R}^p$ is the center of the $k$-th cluster and $\epsilon_i$ is an i.i.d. random noise term following the common distribution $N(0, \Sigma_k)$. Here we focus on the most general and realistic scenario where the within-cluster covariance matrices $\Sigma_1, \ldots, \Sigma_K$ are heterogeneous. In our formulation of the GMM, the true partition $(G_k^*)_{k=1}^K$ is treated as a *unknown* parameter in model (20), along with the component-wise parameters $(\mu_k, \Sigma_k)_{k=1}^K$. With this parameterization $(G_k, \mu_k, \Sigma_k)_{k=1}^K$, the log-likelihood function for observing the data $\mathbf{X} = \{X_1, \ldots, X_n\}$ is given by

$$\ell\big((G_k, \mu_k, \Sigma_k)_{k=1}^K \mid \mathbf{X}\big) = -\sum_{k=1}^K \frac{|G_k|}{2} \log(2\pi|\Sigma_k|) - \frac{1}{2} \sum_{k=1}^K \sum_{i \in G_k} (X_i - \mu_k)^T \Sigma_k^{-1} (X_i - \mu_k),$$

where $|G_k|$ is the cardinality of $G_k$ and $|\Sigma_k|$ is the determinant of matrix $\Sigma_k$. Since we are primarily interested in recovering the clustering labels (or equivalently the assignment matrix, cf. Section 3.1 below) in the presence of cluster heterogeneity, we can first profile out the nuisance parameters $\mu_k$ in closed form and the resulting objective function as a profile log-likelihood for the remaining parameters (after dropping constants) is given by

$$\ell\big((G_k, \Sigma_k)_{k=1}^K \mid \mathbf{X}\big) = -\sum_{k=1}^K |G_k| \log(|\Sigma_k|) - \sum_{k=1}^K \sum_{i \in G_k} \|X_i\|^2_{\Sigma_k^{-1}} + \sum_{k=1}^K \frac{1}{|G_k|} \sum_{i,j \in G_k} \langle X_i, X_j \rangle_{\Sigma_k^{-1}},$$

(2)

where $\langle v, u \rangle_\Sigma := v^T \Sigma u$ and $\|u\|^2_\Sigma := \langle u, u \rangle_\Sigma$ for any $u, v \in \mathbb{R}^p$ and $\Sigma \succ 0$. This leads us to a combinatorial optimization problem for the profile log-likelihood function

$$\max \left\{ \ell\big((G_k, \Sigma_k)_{k=1}^K \mid \mathbf{X}\big) : \bigsqcup_{k=1}^K G_k = [n], \; \Sigma_k \succ 0 \right\},$$

(3)

where the disjoint union $\bigsqcup_{k=1}^K G_k = [n]$ means that $\bigcup_{k=1}^K G_k = [n]$ and $G_j \cap G_k = \emptyset$ if $j \neq k$. Note that the constrained optimization problem in (3) in the special case $\Sigma_1 = \cdots = \Sigma_k = \sigma^2 \mathrm{Id}_p$ reduces to the $K$-means clustering method, which is known to be worst-case NP-hard (Dasgupta, 2007; Mahajan et al., 2009). To overcome such computational difficulty, semidefinite program (SDP) relaxation is a tractable solution that achieves information-theoretically optimal exact recovery under the standard GMM with identical and isotropic covariance matrices (Chen & Yang, 2021a). Nevertheless, all existing formulations of various SDP relaxations of the standard GMM critically depend on the assumption that $\Sigma_1 = \cdots = \Sigma_k = \sigma^2 \mathrm{Id}_p$ with a *known* noise variance parameter $\sigma^2$ (Fei & Chen, 2018; Li et al., 2017; Peng & Wei, 2007; Chen & Yang, 2021a). This motivates us to seek alternative SDP formulations adjusting the (full) information coming from the likelihood function for the observed data $\mathbf{X}$.

## 3 LIKELIHOOD ADJUSTED SDP FOR CLUSTERING HETEROGENEOUS DATA

In this section, we introduce the likelihood adjusted SDP (LA-SDP) for recovering the true partition structure $G_1^*, \ldots, G_K^*$ by applying convex relaxation to the profile log-likelihood function (3).

### 3.1 ORACLE LA-SDP UNDER KNOWN COVARIANCE MATRICES

In this subsection, we consider the oracle case where the covariance matrices $\Sigma_1, \ldots, \Sigma_K$ are known. Let us start with a well-studied SDP relaxation formulation (Peng & Wei, 2007) for approximating the combinatorial optimization problem of maximizing the profile log-likelihood function under the isotropic setting with known $\Sigma_1 = \ldots = \Sigma_K = \sigma^2 \mathrm{Id}_p$, which is known (Chen & Yang, 2021b) to attain the information-theoretically optimal threshold on centroid separations for exact recovery of cluster labels. Note that there is a one-to-one correspondence between any given partition $(G_k)_{k=1}^K$ of $[n]$ and a binary assignment matrix $H = (h_{ik}) \in \{0,1\}^{n \times K}$ (up to cluster labels permutation) such that $h_{ik} = 1$ if $i \in G_k$ and $h_{ik} = 0$ otherwise for $i \in [n]$ and $k \in [K]$. Because each row of $H$ contains exactly one non-zero entry, the recovery of the true clustering structure (or its associated assignment matrix) by maximizing the profile log-likelihood function (after dropping constants) can be re-expressed as a (non-convex) mixed integer program:

$$\max_H \langle A, HBH^\top \rangle = \sum_{k=1}^K \frac{1}{|G_k|} \sum_{i,j \in G_k} \langle X_i, X_j \rangle, \quad \text{subject to } H \in \{0,1\}^{n \times K} \text{ and } H\mathbf{1}_K = \mathbf{1}_n, \quad (4)$$

where $A = X^\top X$ is the $n \times n$ similarity matrix, $\mathbf{1}_n$ denotes the $n$-dimensional vector of all ones, and $B$ is the $K \times K$ diagonal matrix whose $k$-th diagonal component is $|G_k|^{-1} = \big(\sum_{i=1}^n h_{ik}\big)^{-1}$. Here, we have used the key identity $\sum_{k=1}^K w_k \sum_{i,j \in G_k} a_{ij} = \langle A, HBH^\top \rangle$ that holds for any diagonal matrix $B = \mathrm{diag}(w_1, \ldots, w_K)$ and similarity matrix $A = (a_{ij})_{i,j=1}^n$. Relaxing the above mixed integer program (4) by lifting the assignment matrix $H$ into $Z = HBH^\top$, we arrive at its SDP relaxation as

$$\hat{Z} = \arg \max_{Z \in \mathbb{R}^{n \times n}} \langle A, Z \rangle, \quad \text{subject to } Z \succeq 0, \; \mathrm{tr}(Z) = K, \; Z\mathbf{1}_n = \mathbf{1}_n, \; Z \geqslant 0, \quad (5)$$

where $Z \geqslant 0$ means each entry $Z_{ij} \geqslant 0$ and $Z \succeq 0$ means the matrix $Z$ is symmetric and positive semi-definite. This SDP formulation relaxes the integer constraint on $H$ into two linear constraints $\mathrm{tr}(Z) = K$ and $Z \geqslant 0$ that are satisfied by any $Z = HBH^T$ as $H$ ranges over feasible solutions of problem (4).

Now let us consider the general heterogeneous setting with (possibly) different and non-isotropic covariance matrices $\Sigma_1, \ldots, \Sigma_K$, and extend the SDP relaxation to this setting. Two technical difficulties arise by examining the previous argument. First, the first two terms in the profile log-likelihood function (3) are no longer independent of the assignment matrix, and is therefore not negligible. In particular, they also provide partial information about the cluster labels when the covariance matrices are different: $\|X_i\|^2_{\Sigma_k^{-1}}$ in the second term quantifies how well $X_i$ aligns with the covariance matrix $\Sigma_k$ encoding second-order information of the $k$-th cluster; while the first term plays the role of balancing the cluster sizes and favors assigning more points to clusters with smaller shapes (since density is expected to be high). Second, the similarity $\langle X_i, X_j \rangle_{\Sigma_k^{-1}}$ within cluster $G_k$ in the third term now depends on $k$, making the key identity $\sum_{k=1}^K w_k \sum_{i,j \in G_k} a_{ij} = \langle A, HBH^\top \rangle$ for connecting the profile log-likelihood function with the objective function of the mixed integer program (4) no longer applicable.

To solve the two aforementioned difficulties, we propose to augment the single variable $Z$ in the SDP relaxation (5) to $K$ variables $(Z_k)_{k=1}^K$, where $Z_k$ can be interpreted as the lifting of the $k$-th column $H_k$ of the assignment matrix $H$ via $Z_k = \frac{1}{|G_k|} H_k H_k^\top$, $|G_k| = \sum_{i=1}^n h_{ik} = H_k^\top \mathbf{1}_n$, that encodes cluster membership associated with the $k$-th cluster. More specifically, by extending the key identity in the isotropic setting to $\sum_{k=1}^K w_k \sum_{i,j \in G_k} a_{ij}^{(k)} = \sum_{k=1}^K \langle A^{(k)}, H_k w_k H_k^\top \rangle$ for any weight vector $w = (w_k)_{k=1}^K$ and $K$ similarity matrices $\left(A_k = (a_{ij}^{(k)})_{i,j=1}^n\right)_{k=1}^K$, we can analogously express the maximizing profile log-likelihood problem as the following (non-convex) mixed integer program:

$$\max_H \sum_{k=1}^K \langle A^{(k)}, H_k w_k H_k^\top \rangle, \quad \text{subject to } H_k \in \{0,1\}^{n \times 1} \text{ and } \sum_{k=1}^K H_k = \mathbf{1}_n, \qquad (6)$$

where $w_k = |G_k|^{-1} = \left(\sum_{i=1}^n h_{ik}\right)^{-1}$, and the $k$-th cluster-specific similarity matrix $A^{(k)}$ is

$$A^{(k)} := -\log(|\Sigma_k|) \mathbf{1}_n \mathbf{1}_n^T - \frac{1}{2} \left[\mathrm{diag}(X^T \Sigma_k^{-1} X) \mathbf{1}_n^T + \mathbf{1}_n \mathrm{diag}(X^T \Sigma_k^{-1} X)^T \right] + X^T \Sigma_k^{-1} X. \quad (7)$$

Here, $\mathrm{diag}(A)$ stands for the column vector composed of all diagonal entries of a matrix $A$. Now by lifting $H_k$ into $Z_k = H_k w_k H_k^\top$, we arrive at the following SDP relaxation for the profile log-likelihood objective function (2):

$$\left(\hat{Z}_1, \ldots, \hat{Z}_K\right) = \operatorname*{arg\,max}_{Z_1, \ldots Z_K \in \mathbb{R}^{n \times n}} \sum_{k=1}^K \langle A_k, Z_k \rangle,$$

$$\text{subject to } Z_k \succeq 0, \ \sum_{k=1}^K \mathrm{tr}(Z_k) = K, \ \left(\sum_{k=1}^K Z_k\right) \mathbf{1}_n = \mathbf{1}_n, \ Z_k \geqslant 0, \ \forall\, k \in [K], \qquad (8)$$

which relaxes the integer constraint on $H = (H_1, H_2, \cdots, H_k)$ into $(K+1)$ linear constraints $\sum_{k=1}^K \mathrm{tr}(Z_k) = K$ and $Z_k \geqslant 0$ for $k \in [K]$ that are satisfied by any $Z_k = H_k w_k H_k^\top$ as $H$ ranges over feasible solutions of problem (6).

Since solving (8) requires the knowledge of the true covariance matrix for each component, we call the solution $(\hat{Z}_k)_{k=1}^K$ as the *oracle* likelihood adjusted SDP (LA-SDP) for estimating the cluster membership matrix of data points. In the special case of isotropic covariance matrices $\Sigma_1 = \cdots = \Sigma_K = \sigma^2 \mathrm{Id}_p$, Proposition 1 below shows that LA-SDP reduces to become equivalent to the previous SDP formulation (5).

*Proposition* 1 (**SDP relaxation for $K$-means is a special case of LA-SDP**). Suppose $\Sigma_k = \sigma^2 \mathrm{Id}_p$ for all $k \in [K]$. Let $\hat{Z}$ be the solution to (5) that achieves maximum $M_1$ and $\hat{Z}_k, k = 1, \ldots, K$, be the solution to (5) with maximum $M_2$. Then $M_1 = M_2$. And $\hat{Z} = \sum_{k=1}^K \hat{Z}_k$, if $\hat{Z}$ is unique in (5).

Note that the SDP relaxed $K$-means in (5) is originally proposed in (Peng & Wei, 2007) and has been extensively studied in literature. In particular, it achieves the information-theoretical limit for exact recovery under the standard GMM (Chen & Yang, 2021a) and it is robust against outliers and adversarial attack (Fei & Chen, 2018). In the case of exact recovery where $\hat{Z} = Z^*$ and $Z^*$ is the true cluster membership matrix such that $Z^*_{ij} = |G^*_k|^{-1}$ if $i, j \in G^*_k$ and $Z^*_{ij} = 0$ otherwise, then we can easily recover the true partition structure $G^*_1, \ldots, G^*_K$ or its associated assignment matrix from the block diagonal matrix $\hat{Z}$.

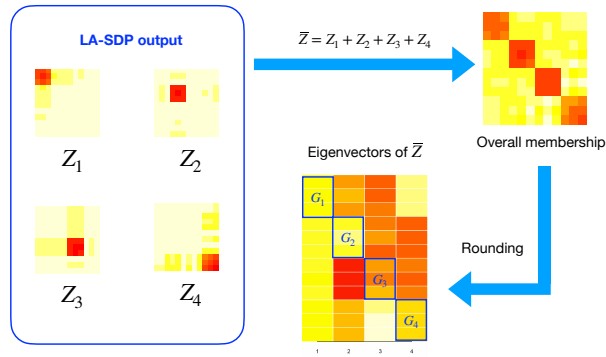

Figure 1: LA-SDP membership matrices to cluster labels via spectral rounding.

Thus it is an interesting theoretical question of when the partition structure induced by $\hat{Z} = \sum_{k=1}^K \hat{Z}_k$ from the LA-SDP (see Figure 1 for an illustration) can achieve exact recovery. Theorem 2 below gives a lower bound of the separation signal-to-noise ratio for achieving exact recovery in the presence of data heterogeneity.

For each distinct pair $(k, l) \in [K]$, let $D_{(k,l)} := \frac{\sum_{i=1}^p (\lambda_i - \log(1+\lambda_i))}{p \max_i |\lambda_i|}$ characterize the closeness between $\Sigma_k$ and $\Sigma_l$, where $\lambda_1, \ldots, \lambda_p$ enumerate all eigenvalues of $(\Sigma_l^{1/2} \Sigma_k^{-1} \Sigma_l^{1/2} - \mathrm{Id}_p)$. If $\lambda_i = 0, \ \forall i \in [p]$, we let $D_{(k,l)} = 0$. Let $\Delta^2 := \min_{k \neq l} \|\Sigma_k^{-1/2}(\mu_k - \mu_l)\|^2$ denote a covariance adjusted centroid separation, $n_k := |G^*_k|$ the size of true cluster $G^*_k$, $m = \min_{k \neq l} \frac{2n_k n_l}{n_k + n_l}$ the least pairwise harmonic mean over cluster sizes, $\underline{n} = \min_k n_k$ the minimal cluster size, and $M := \max_{k \neq l} \|\Sigma_l^{1/2} \Sigma_k^{-1} \Sigma_l^{1/2}\|_{\mathrm{op}}$ (matrix operator norm).

*Theorem* 2 (**Exact recovery for LA-SDP**). Suppose there exist constants $\delta > 0$, $\beta \in (0, 1)$ and $\eta \in (0, 1)$ such that

$$\log n \geq \max\left\{ \frac{(1-\beta)^2}{\beta^2}, \frac{(1-\beta)(1-\eta)K^2}{\beta^2 \max\{(M-1)^2, 1\}} \right\} \frac{C_1 n}{m}, \ \delta \leq \frac{\beta^2}{(1-\beta)^2} \frac{C_2 M^{1/2}}{K}, \ m \geq \frac{4(1+\delta)^2}{\delta^2}.$$

Then the LA-SDP achieves exact recovery, or $\hat{Z} = Z^*$, with probability at least $1 - C_7 K^3 n^{-\delta}$ if

$$\Delta^2 \geq (E_1 + E_2) \log n, \text{ and } \min_{k \neq l} D_{(k,l)} \geq C_3(1 + \log n/p + p/n), \tag{9}$$

where concrete expressions of $E_1$ and $E_2$ (depending on $\delta, \beta, \eta$) are provided in Appendix A.4, and $C_1, \ldots, C_7$ are universal constants.

Our definition of the centroid separation $\Delta$ extends the separation-to-noise ratio (SNR) for the exact recovery under the isotropic covariance setting (Chen & Yang, 2021a) to the heterogeneous setting by taking into account the cluster shapes (i.e. second order information). From (18), we see that our theoretical centroid separation lower bound consists of two parts $E_1$ and $E_2$: $E_1$ reduced to the information-theoretically optimal threshold when $M = 1$, corresponding to same covariance matrices; $E_2$ tends to vanish for small $M$ close to one and satisfying $M = 1 + o(1/\sqrt{n \log n})$ or remains as an extra term for large $M$. From our numerical results summarized in Figure 2, we can observe that our defined centroid separation $\Delta$ indeed captures the accuracy of cluster label recovery using LA-SDP—the mis-clustering error curves display almost identical patterns under different settings of the GMM. In comparison, the performance of the (original) SDP (5) and the $K$-means clustering method designed for the isotropic case become significantly worse as the condition number of the cluster covariance matrices increases. More details about implementation and model setups are provided in Appendix A.3.

### 3.2 ITERATIVE LA-SDP UNDER UNKNOWN COVARIANCE MATRICES: AN ALTERNATING MAXIMIZATION ALGORITHM

Since the oracle LA-SDP relies on the knowledge of covariance matrices $\Sigma_1, \ldots, \Sigma_K$, we propose a simple and practical data-driven algorithm for approximating LA-SDP when these covariance

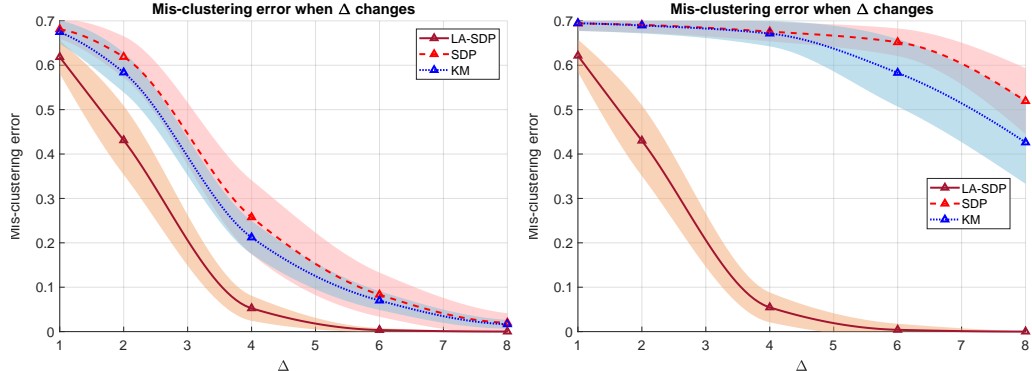

Figure 2: Mis-clustering error (with shaded error bars) vs centroid separation $\Delta$ under different conditional numbers of cluster covariance matrices $\Sigma_1 = \Sigma_2 = \cdots = \Sigma_K$ ($M = 1$). The left (right) plot corresponds to a moderate (large) condition number of the common covariance matrix. Here, KM refers to $K$-means method; SDP refers to the original SDP (5).

matrices are unknown. The idea is to alternate between the SDP relaxation given a current estimate of the component covariance matrices and updating covariance matrices according to the maximum (penalized) likelihood given the new membership estimate. The next lemma gives a closed-form formula for updating covariance matrices given a current estimate of the assignments $Z_1, \ldots, Z_K$ based on their (unconstrained) MLEs on the observed data.

*Lemma* 3 (**Updating formula for covariance matrices under alternating maximization**). For any feasible matrices $Z_1, \ldots, Z_K$ satisfying the constraints of (8),

$$\hat{\Sigma}_k := \frac{1}{\mathbf{1}_n^T Z_k \mathbf{1}_n} \sum_{i,j=1}^n \left[ \frac{1}{2}(X_i X_i^T + X_j X_j^T) - X_i X_j^T \right] Z_{k,ij}, \quad k \in [K], \tag{10}$$

solve the following optimization problem

$$\hat{\Sigma}_1, \ldots, \hat{\Sigma}_K = \arg \max_{\Sigma_1, \ldots \Sigma_K \succeq 0} \sum_{k=1}^K \langle A_k, Z_k \rangle, \tag{11}$$

where recall that $A^{(k)} := A^{(k)}(\Sigma_k)$ is the $\Sigma_k$-dependent similarity matrix defined in (7).

Based on the lemma, we propose an *iterative LA-SDP* (iLA-SDP) by alternating maximization of the profile log-likelihood (3) for estimating the lifted cluster membership matrices $(Z_k)_{k=1}^K$ from LA-SDP (8) and the component covariance matrices $(\Sigma_k)_{k=1}^K$, as summarized in Algorithm 1. In the special case where the lifted membership matrix $Z_k$ is of rank one, which holds for true lifted cluster membership matrices $(Z_k^*)_{k=1}^K$, the covariance matrices produced by iLA-SDP can be interpreted as within-cluster sample covariance matrices under soft clustering.

*Proposition* 4 (**Covariance estimation in iLA-SDP via soft clustering**). If $\text{rank}(Z_k) = 1$, then there exists weights $(w_{k,1}, \ldots, w_{k,n})$ such that these $\hat{\Sigma}_k$ in Lemma 3 can be written as

$$\hat{\Sigma}_k := \frac{1}{n_k} \sum_{i=1}^n w_{k,i}(X_i - \hat{\mu}_k)(X_i - \hat{\mu}_k)^\top, \text{ where } \hat{\mu}_k := \frac{1}{n_k} \sum_{i=1}^n w_{k,i} X_i \text{ and } n_k = \sum_{i=1}^n w_{k,i}. \tag{12}$$

It is further noted from the proof of Proposition 4 that when $Z_k$ has rank one, the weights $w_{k,1}, \ldots, w_{k,n}$ are proportional to the leading non-zero eigenvector of $Z_k$. Thus the alternating maximization step (17) for updating the covariance matrices in iLA-SDP can be interpreted as a soft clustering technique that resembles the EM algorithm. Specifically, the E-step estimates the (hard) cluster label $Y_i \in \{0, 1\}^K$ associated with $X_i$ by the posterior probabilities $\tau_{ik} := p(Y_{ik} \mid X_i, \hat{\theta}^{(t)})$ where $\hat{\theta}^{(t)} = (\hat{\pi}_k^{(t)}, \hat{\mu}_k^{(t)}, \hat{\Sigma}_k^{(t)})_{k=1}^K$ denotes the estimated GMM parameters at the $t$-th iteration in the EM. Then the M-step updates the parameters via $\hat{\pi}_k^{(t+1)} = m_k^{(t)}/n$ with $m_k^{(t)} = \sum_{i=1}^n \tau_{ik}^{(t)}$,

$$\hat{\mu}_k^{(t+1)} = \frac{1}{m_k^{(t)}} \sum_{i=1}^n \tau_{ik}^{(t)} X_i \text{ and } \hat{\Sigma}_k^{(t+1)} = \frac{1}{m_k^{(t)}} \sum_{i=1}^n \tau_{ik}^{(t)}(X_i - \hat{\mu}_k^{(t+1)})(X_i - \hat{\mu}_k^{(t+1)})^\top. \tag{13}$$

Note that (17) and (13) represent different weighting schemes in the soft clustering rule for obtaining an estimate for the cluster labels. In iLA-SDP, the weight $w_{k,i}$ for $X_i$ belonging to component $k$ is determined by the SDP in (8). Once the weights are calculated, remaining parameter updates in both iLA-SDP and EM boil down to simple averages with effective component sample sizes $n_k$ and $m_k$, respectively. In Section 3.3 to follow, we provide deeper comparison between iLA-SDP and EM.

*Remark* 5. In Appendix A.2, we further propose two variations of iLA-SDP that can handle high-dimensional and large-size data with better computational and statistical efficiency. For high-dimensional data, we apply Fisher's LDA with an initial estimate of the cluster labels to find an optimal feature subspace that increases the SNR for better clustering, and for large-size data we combine the subsampling idea with iLA-SDP to reduce computational cost (Zhuang et al., 2022b).

---

**Algorithm 1:** The iterative likelihood adjusted SDP (iLA-SDP) algorithm

**Input:** Data matrix $X \in \mathbb{R}^{p \times n}$ containing $n$ points. Initialization of assignments $G_1^{(0)}, \ldots, G_K^{(0)}$ or covariance matrices $\Sigma_1^{(0)}, \ldots, \Sigma_K^{(0)}$. The stopping criterion parameters $\epsilon$, $S$.

1 (Assignments to covariance matrices) If we have the initialization of assignments, let
$\Sigma_k^{(0)} := |G_k^{(0)}|^{-1} \sum_{i \in G_k^{(0)}} (X_i - \bar{X}_k)(X_i - \bar{X}_k)^T$ to be the sample covariance of each cluster $k \in [K]$, where $\bar{X}_k := |G_k^{(0)}|^{-1} \sum_{i \in G_k^{(0)}} X_i$.

2 **for** $s = 1, \ldots, S$ **do**

3     (Adjusted-SDP) Solve the Adjusted-SDP in (8) using $X$ and $\Sigma_1^{(s-1)}, \ldots, \Sigma_K^{(s-1)}$ to get solution $Z_1^{(s)}, \ldots, Z_K^{(s)}$.

4     Compute the sum $\bar{Z}^{(s)} := \sum_{k=1}^K Z_k^{(s)}$ and the relative norm
    $r^{(s)} := \|\bar{Z}^{(s)} - \bar{Z}^{(s-1)}\|_F / \|\bar{Z}^{(s-1)}\|_F$ for $s \geq 2$. We will break the loop if $r^{(s)} < \epsilon$.

5     (Assignments to covariance matrices) Use formula in Lemma 3 to get covariance matrices
    $\Sigma_1^{(s)}, \ldots, \Sigma_K^{(s)}$ from $Z_1^{(s)}, \ldots, Z_K^{(s)}$.

6 Perform the spectral decomposition of $\bar{Z}^{(S)}$ and take the top $K$ eigenvectors $(\hat{u}_1, \ldots, \hat{u}_K)$.

7 Run $K$-means clustering on $(\hat{u}_1, \ldots, \hat{u}_K)$ and extract the cluster labels $\hat{G}_1, \ldots, \hat{G}_K$ as a partition estimate for $[n]$.

**Output:** A partition estimate $\hat{G}_1, \ldots, \hat{G}_K$ for $[n]$.

---

### 3.3 CONNECTIONS BETWEEN iLA-SDP AND EM ALGORITHMS

It is interesting to observe that our proposed iLA-SDP algorithm is closely connected to the classic EM algorithm, which approximates the maximum likelihood estimation (MLE) of the observed data in statistical models with latent variables (Dempster et al., 1977). The key idea of EM algorithm in the model-based clustering context is *data augmentation* where the latent variables represent the cluster labels. More specifically, for each data point $X_i \in \mathbb{R}^p$, we associate with an unobserved one-hot encoded cluster label $Y_i := \{Y_{i1}, \ldots, Y_{iK}\} \in \{0, 1\}^K$. Then the EM algorithm aims to iteratively maximize the *expected log-likelihood of the complete data* $(X_i, Y_i)_{i=1}^n$ given by

$$\theta^{(t+1)} = \arg\max_\theta \left\{ Q(\theta \mid \theta^{(t)}) := \mathbb{E}_{\mathbf{Y} \sim q(\cdot \mid \mathbf{X}, \theta^{(t)})}[\ell_c(\theta \mid \mathbf{X}, \mathbf{Y})] \right\}, \tag{14}$$

where $\theta = ((\pi_k, \mu_k, \Sigma_k)_{k=1}^K)$ contains parameters in the GMM, $(\pi_k)_{k=1}^K$ are the weight parameters such that $\pi_k \geqslant 0$ and $\sum_{k=1}^K \pi_k = 1$, and the complete log-likelihood function is

$$\ell_c(\theta \mid \mathbf{X}, \mathbf{Y}) := p(\mathbf{X}, \mathbf{Y} \mid \theta) = -\frac{1}{2} \sum_{i=1}^n \sum_{k=1}^K Y_{ik} \left[ \log(2\pi|\Sigma_k|) - (X_i - \mu_k)^\top \Sigma_k^{-1}(X_i - \mu_k) \right].$$

Alternatively, the EM algorithm (14) can be interpreted as minorize-maximization (MM) that maximizes a best lower bound for the log-likelihood of the observed data

$$\ell(\theta \mid \mathbf{X}) := \log p(\mathbf{X} \mid \theta) = \log \sum_{\mathbf{Y}} p(\mathbf{X}, \mathbf{Y} \mid \theta) \geqslant \sum_{\mathbf{Y}} q(\mathbf{Y} \mid \mathbf{X}) \log \frac{p(\mathbf{X}, \mathbf{Y} \mid \theta)}{q(\mathbf{Y} \mid \mathbf{X})} =: \mathcal{L}(q, \theta)$$

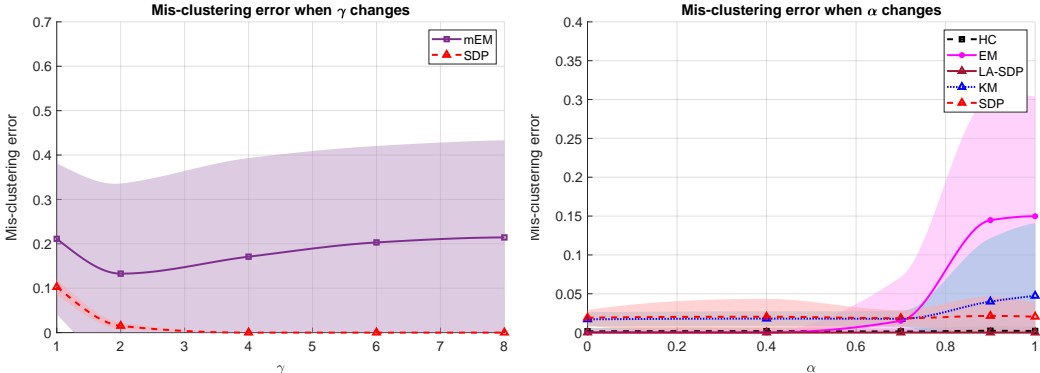

Figure 3: Mis-clustering error (with shaded error bars) vs $\gamma$ (captures the signal strength of GMM) and $\alpha$ (perturbation percentage of initialization). mEM (SDP) refers to the reduced version of EM (LA-SDP) where we consider covariance matrices as fixed and equal to identity. The first plot compares the performance of mEM and SDP when separation is large with random initialization; the second plot compares all methods when we enlarge the perturbation percentage $\alpha$ applied to the random initialization from hierarchical clustering (HC).

for any posterior distribution $q(\mathbf{Y} \mid \mathbf{X})$. Under this perspective, the EM algorithm can be expressed as an *alternating maximization* algorithm on $\mathcal{L}(q, \theta)$ between E-step $q^{(t+1)} = \arg\max_q \mathcal{L}(q, \theta^{(t)})$ and M-step $\theta^{(t+1)} = \arg\max_\theta \mathcal{L}(q^{(t+1)}, \theta)$. Thus, give any $q(\mathbf{Y} \mid \mathbf{X})$, the M-step maximizes the expected complete log-likelihood as a surrogate function that minorizes $\ell(\theta \mid \mathbf{X})$ because $\mathcal{L}(q, \theta) = \mathbb{E}_{\mathbf{Y} \sim q(\cdot \mid \mathbf{X})}[\ell_c(\theta \mid \mathbf{X}, \mathbf{Y})] - H(q(\mathbf{Y} \mid \mathbf{X}))$ where $H(q)$ denotes the relative entropy of distribution $q$, while given the current parameter estimate $\theta^{(t)}$, the E-step is maximized at $q^{(t+1)}(\mathbf{Y} \mid \mathbf{X}) = p(\mathbf{Y} \mid \mathbf{X}, \theta^{(t)})$ because

$$\ell(\theta^{(t)} \mid \mathbf{X}) \geqslant \mathcal{L}(p(\mathbf{Y} \mid \mathbf{X}, \theta^{(t)}), \theta^{(t)}) = \sum_{\mathbf{Y}} p(\mathbf{Y} \mid \mathbf{X}, \theta^{(t)}) \log p(\mathbf{X} \mid \theta^{(t)}) = \ell(\theta^{(t)} \mid \mathbf{X}),$$

where the first inequality is actually an equality at $p(\mathbf{Y} \mid \mathbf{X}, \theta^{(t)})$. Even though the EM and iLA-SDP are both alternating maximization algorithms aiming to solve the MLE for the observed data log-likelihood and both can be viewed as soft clustering methods (cf. Proposition 4), there are several important differences we would like to highlight.

First, cluster labels are (random) latent variables and they are estimated via posterior probabilities in the EM algorithm, while the labels are treated as unknown parameters in iLA-SDP that are estimated via direct maximization of the observed data likelihood.

Second, the EM algorithm is a special case of the minorization-maximization (MM) algorithm (Hunter & Lange, 2000) by iteratively performing the coordinate ascent on the expected complete data log-likelihood as a minorizing surrogate function, while our iLA-SDP is *exact* in the sense that it directly optimizes the observed data log-likelihood via a convex relaxation formulation. Thus iLA-SDP is a more direct approach than EM for tackling the non-convex observed log-likelihood objective and it is principled to perfectly recover the true clustering structure if the clusters are well-separated under an SNR lower bound in Theorem 2. As in the EM algorithm, iLA-SDP monotonically maximizes the observed data log-likelihood over iterations; cf. Figure 7 in Appendix.

Third, the EM algorithm in each iteration must estimate the cluster center parameters $(\mu_k)_{k=1}^{K}$, while our iLA-SDP profiles out the effect of centroid estimation and leverages only pairwise Mahalanobis distances between data to accommodate the heterogeneity of cluster shapes. Partly because the error in estimating the centroids propagates to other parameters, EM is more sensitive to initialization with inaccurate labels and the centroid configurations even in the standard GMM (Jin et al., 2016), and iLA-SDP behaves better than EM, an observation we empirically verify in our simulation experiments; cf. Figure 3 for comparison between iLA-SPD and EM algorithms. From the first plot we can observe that LA-SDP with isotropic known covariance matrices, which reduces to the $K$-means SDP in (5), performs stable and achieves exact recovery when the separation is large. However, EM fails with random initialization in this adversarial centroids configuration. Moreover, from the second figure we can see that LA-SDP is fairly stable with perturbation of initialization if the separation is large while EM can go worse as the perturbation percentage of initialization $\alpha$ approaches 1, i.e., all the labels

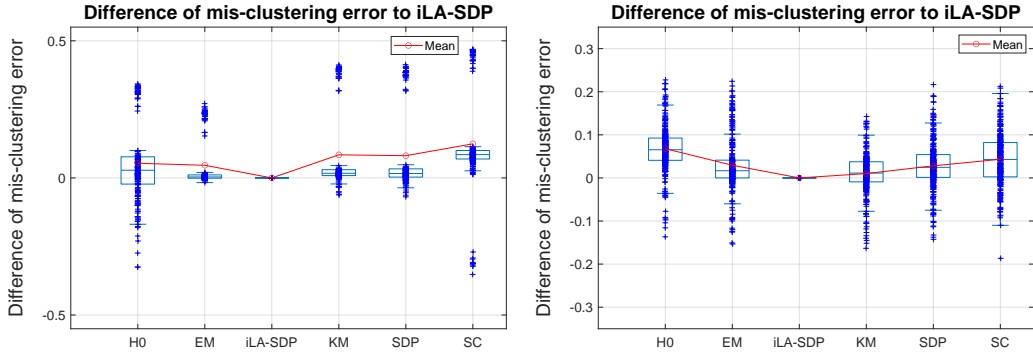

Figure 4: Box plots of difference of mis-clustering error (with means) for different methods to iLA-SDP. The left (right) plot summarizes the results for the banknote authentication dataset (landsat satellite dataset). Here SC refers to spectral clustering method.

are selected randomly. In other words, EM is more sensitive to initialization and iLA-SDP is more stable if the signal is strong. More details of the settings in Figure 3 can be found in Appendix A.3.

## 4   REAL-DATA APPLICATIONS

In this section, we test the performance of iLA-SDP against several widely used clustering methods on two real datasets from the UCI machine learning repository.

**Banknote authentication dataset.** We first look at the performances of our methods for a banknote authentication dataset where the separation of clusters are not large. The images were taken from genuine and forged banknote-like specimens, where the features were extracted by Wavelet Transform tool. It contains 1372 samples and $p = 4$ attributes with $K = 2$ clusters. We choose total $n = 1000$ samples randomly and equally from two clusters to make cluster sizes balanced among total 200 replicates. HC is used as initialization for EM, KM and iLA-SDP. If the initialization for the assignments $G_1^{(0)}$, $G_2^{(0)}$ is highly unbalanced, i.e., $\|G_1^{(0)}\|/\|G_2^{(0)}\| > 4$ if $\|G_1^{(0)}\| > \|G_2^{(0)}\|$, then the covariance matrices of two clusters should differ significantly and we calculate the covariance estimation for unconstrained optimization problem; Otherwise we will calculate the estimation of the covariance matrices through graphical lasso with parameter $\lambda = 2$ since the similarity shows that we could reduce the estimation of parameters. Then we run Algorithm 1 to get the results for iLA-SDP. The comparison of those four methods can be found from the left plot in Figure 4, where we can observe that the separation between two clusters is not well in the sense that the medians of all methods are similar. Nevertheless, iLA-SDP can achieve better performance than other methods for most of the time. The reason iLA-SDP has lower mean is that sometimes iLA-SDP can achieve nearly exact recovery in the sense that there are 36 out of 200 times when the mis-clustering error for iLA-SDP are below 0.05, while EM can only hit around 0.25. This indicates that there are more chances for iLA-SDP to sort two clusters with fairly good performance even the separation is problematic.

**Landsat satellite dataset.** This database was generated from landsat Multi-Spectral Scanner image data. The test set includes 2000 satellite images, 6 different clusters with 36 attributes ($36 = 4$ spectral bands $\times$ 9 pixels in neighbourhood). Every attribute is an integer from 0 to 255 indicating the color for certain pixel. We performed 4 methods on the transformed dataset with total 300 replicates. For each repetition, we draw total $n = 1200$ samples randomly and equally from clusters to make cluster sizes balanced. And for each attribute, we scale its range to $[0, 1]$ and then take the function $f(x) = \log(1/x - 1)$ entry-wise to transform the range to $\mathbb{R}_+$. Then, we run Algorithm 3 on the transformed dataset $\tilde{X}$ to get the results for iLA-SDP with $\epsilon = 10^{-2}$, $p_0 = K = 6$, and $R = 50$. From the results we can see that our method iLA-SDP performs the best for all four methods. Especially, iLA-SDP out-performs EM since the initialization (HC) is rough, which results in both biases of the estimations of group means and covariance matrices for EM while iLA-SDP only uses the group covariance matrices as its initialization.

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

# A APPENDIX

## A.1 FURTHER RESPONSE TO REVIEWERS

**Sample complexity bound.** To verify the sample complexity bound for LA-SDP in Theorem 2 ($O(\log(n))$) is tight, we will change $n$ and adjust the squared distance between clusters by multiplying $\log(n)$. More precisely, we let $d = \lambda\sqrt{\log(n)}$, $\lambda > 0$. The diagonal of the covariance matrices are placed at a simplex of $\mathbb{R}^p$ that are not identical to the corresponding centers. i.e. $\mu_k = \lambda \cdot e_k$, $\Sigma_k = L \cdot \text{diag}(e_{k+1})$, $\forall l \in [K]$, where $e_{K+1} = e_1$. This guarantees the symmetry of the construction. We set $L = 10$, $p = 4$, $K = 4$. Each time we draw the $n = 120/240/480$ data from the GMM. The results of the simulation for the second plot in Figure 5 are obtained through 20 total replicates, where we can observe the same pattern across different settings for $n$. This shows that the order $\log(n)$ for separation bound in Theorem 2 should be tight.

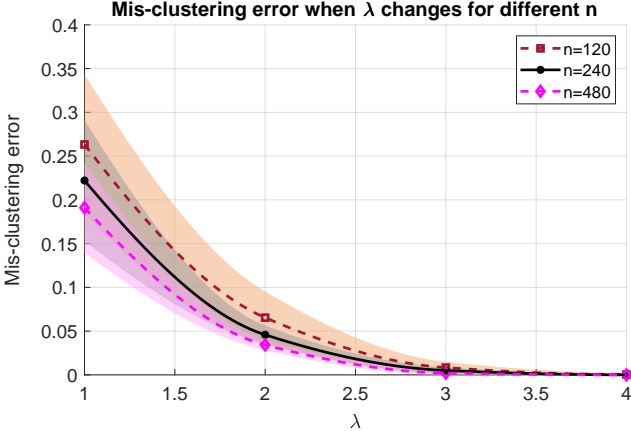

Figure 5: Mis-clustering error (with shaded error bars for the left plot) vs $\lambda$ for iLA-SDP for different $n$.

**Computational complexity for banknote authentication dataset.** Now if we look at the results of time cost for clustering banknote authentication dataset in Table 1, we can observe that the time cost for iLA-SDP is relatively high and to reduce the time cost, we could consider sub-sampling methods, e.g., the subsampling idea (Zhuang et al., 2022b). This will be set as our future goal.

Table 1: Time cost (SD) for clustering banknote authentication dataset for 20 replicates.

| EM | KM | iLA-SDP | SC |
|---|---|---|---|
| 0.1719 (0.0853) | 0.0013 (0.0013) | 2100 (1882) | 0.0395 (0.0959) |

## A.2 ENHANCED iLA-SDPS FOR HIGH-DIMENSIONAL AND LARGE-SIZE DATA

In this section, we propose two variations of iLA-SDP that can handle high-dimensional and large-size data with better computational and statistical efficiency.

**High dimensional data.** If the number attributes of the data are large, it would be hard to approximate the true covariance matrices since there are $O(p^2)$ many unknown parameters. Thus, we propose two dimension reduction procedures that based on hierarchical clustering, Fisher's LDA and F-test. The detailed algorithm have been shown in Algorithm 2 and Algorithm 3. To reduce the dimension, we proposed two procedure.

1. If the number of clusters $K$ is small and the difference between centers are sparse, we shall use HC as a benchmark method for feature selection and assume the group means according to HC

as ground true. Specifically, for $i$-th attribute, we calculate the F-statistics and its p-value based on the $H_0$ that all group means w.r.t. $i$-th attribute are the same. At last, each attribute would likely to be selected if the p-value $\mathcal{P}_i$ for $i$-th attribute is significantly small among p-values for all attributes.

2. First we use the hierarchical clustering to get the clustering results for all possible input cluster number $\tilde{K} \in [p]$. If we assume all the clusters have identical covariance matrices, then we may use the assignments from HC to estimate the within-cluster covariance $\tilde{W}$ (with group means $\tilde{\mu}_l$) and get the signal-to-noise ratio $\Delta(\tilde{K}) := \min_{k \neq l} \|\tilde{W}^{-1/2}(\tilde{\mu}_k - \tilde{\mu}_l)\|$. Here, HC serves as a benchmark method for data initial processing. We will then choose the largest $\tilde{K}$ within target range such that the signal-to-noise ratio $\Delta(\tilde{K})$ is maximized. Then it will lead to the new dataset with dimension $q = \tilde{K} - 1$ after running Fisher's LDA on the assignments from HC with clusters number equals $\tilde{K}$. Finally we perform Algorithm 1 on the new dataset and extract the cluster labels.

**Large-size data.** As we know that the time complexity for solving SDP is as high as $O(n^{3.5})$. We might use subsampling methods to bring down the time cost while maintain the superior behavior for LA-SDP (Zhuang et al., 2022b). The proposed algorithm is shown in Algorithm 4.

---

**Algorithm 2:** Likelihood adjusted SDP based iterative algorithm with unknown covariance matrices $\Sigma_1, \ldots, \Sigma_K$ for large $p$.

**Input:** Data matrix $X \in \mathbb{R}^{p \times n}$ containing $n$ points. Cluster numbers $K$. The stopping criterion parameters $p_0$, $\epsilon$ and $S$. $\alpha \in [0, 1]$, $C > 0$.

1   Run hierarchical clustering with data $X$, clusters number $K$ and extract the cluster labels $G_1^{(0)}, \ldots, G_K^{(0)}$ as prior assignments for $[n]$. Suppose the assignments have true centers $\mu_k^{(0)}$, $k \in [K]$.

2   **for** $i = 1, \ldots, p$ **do**

3      Calculate the p-value $\mathcal{P}_i$ of the F-test $\mathcal{F}_i$ under $H_0$: $\mu_{1,i}^{(0)} = \cdots = \mu_{K,i}^{(0)}$, where $\mu_{k,i}^{(0)}$ corresponds to the $i$-th component of $\mu_i^{(0)}$.

4   Keep $p_0$ attributes with $p_0$ smallest p-values $\mathcal{P}_i$.

5   **if** *there is no clear cutoff between $\mathcal{P}_i$'s, i.e.* $\max_{i \in [p]} \mathcal{P}_i / \min_{i \in [p]} \mathcal{P}_i < C$, **then**

6      we further keep other $p - p_0$ attributes with probability $\alpha > 0$.

7   Get dimension reduced data $\tilde{X}$.

8   Run Algorithm 1 on $\tilde{X}$ with initialization obtained from $K$ clusters of HC and stopping criterion parameters $\epsilon$ and $S$. Then extract the cluster labels $\hat{G}_1, \ldots, \hat{G}_K$ as a partition estimate for $[n]$.

**Output:** A partition estimate $\hat{G}_1, \ldots, \hat{G}_K$ for $[n]$.

---

### A.3 EXPERIMENT RESULTS

In this section, we provide more details of the settings and post the results for simulation experiments. For all the dimension reduction procedures used in the simulation experiments, we perform step 1-7 in Algorithm 2 followed by Algorithm 3 with input parameters $\alpha = 0.7$, $C = 10^{10}$, $p_0 = 2K$, $p_1 = 15$ $\epsilon = 10^{-2}$, $S = 50$. The initialization we use is hierarchical clustering from *mclust* package in R. Here we test our algorithm on Gaussian mixture models and real datasets. We compared our algorithm iLA-SDP (HC as initialization) with HC, EM algorithm (HC as initialization), $K$-means (HC as initialization) and original SDP.

**Improvements of iLA-SDP over SDP.** Recall in Theorem 2, we define the signal-to-noise ratio as $\Delta^2 := \min_{k \neq l} \|\Sigma_k^{-1/2}(\mu_k - \mu_l)\|^2$. To verify the validity of the definition and compare iLA-SDP and SDP, we change the conditional number for covariance matrices $\Sigma_1, \ldots, \Sigma_K$. Here we choose $n = 200$, $p = 4$, $K = 4$. Recall $M := \max_{k \neq l} \|\Sigma_l^{1/2} \Sigma_k^{-1} \Sigma_l^{1/2}\|_{\text{op}}$, we choose all the covariance matrices to be the same such that $M$ is fixed. The covariance matrices are set to be identity matrix except that the first entry at the diagonal are set to be $L + 1$, which refers to the condition number of matrices. We consider two cases where $L = 10, 100$. Now denote $e_k \in \mathbb{R}^p$ as the vector with $k$-th

---

**Algorithm 3:** Likelihood adjusted SDP based iterative algorithm with unknown covariance matrices $\Sigma_1, \ldots, \Sigma_K$ for large $p$.

---

**Input:** Data matrix $X \in \mathbb{R}^{p \times n}$ containing $n$ points. Cluster numbers $K$. The stopping criterion parameters $p_1, \epsilon$ and $S$.

1 Select a bench mark clustering method (HC) as a way to provide a prior assignments.
2 **for** $\tilde{K} = K, K+1, \ldots, p_1 - 1, p_1$ **do**
3    Run hierarchical clustering with data $X$, clusters number $\tilde{K}$ and extract the cluster labels
     $G_1^{(\tilde{K})}, \ldots, G_{\tilde{K}}^{(\tilde{K})}$ as prior assignments for $[n]$ and get the group means $\mu_k^{(\tilde{K})}, k \in [\tilde{K}]$.
4    Calculate the within-cluster covariance matrix $W$, then get the signal-to-noise ratio
     $\Delta(\tilde{K}) := \min_{l \neq k} \|W^{-1/2}(\mu_l^{(\tilde{K})} - \mu_k^{(\tilde{K})})\|$.
5 Choose $K^*$ such that $\Delta(K^*)$ is maximized for $K^* = K, K+1 \ldots, P-1, P$.
6 Perform the Fisher's LDA with data $X$, assignments $G_1^{(K^*)}, \ldots, G_{K^*}^{(K^*)}$ and get the transformed data $\tilde{X} \in \mathbb{R}^{q \times n}$ with $q = K^* - 1$.
7 Run Algorithm 1 on $\tilde{X}$ with initialization obtained from $K$ clusters of HC and stopping criterion parameters $\epsilon$ and $S$. Then extract the cluster labels $\hat{G}_1, \ldots, \hat{G}_K$ as a partition estimate for $[n]$.
**Output:** A partition estimate $\hat{G}_1, \ldots, \hat{G}_K$ for $[n]$.

---

**Algorithm 4:** Sketch and lift: Likelihood adjusted SDP based iterative algorithm with unknown covariance matrices $\Sigma_1, \ldots, \Sigma_K$ for large $n$.

---

**Input:** Data matrix $X \in \mathbb{R}^{p \times n}$ containing $n$ points. Cluster numbers $K$. The stopping criterion parameters $P, \epsilon$ and $S$. Sampling weights $(w_1, \ldots, w_n)$ with $w_1 = \cdots = w_n = \gamma \in (0, 1)$ being the subsampling factor.

1 (Sketch) Independent sample an index subset $T \subset [n]$ via $\text{Ber}(w_i)$ and store the subsampled data matrix $V = (X_i)_{i \in T}$.
2 Run subroutine Algorithm 1 with input $V$ to get a partition estimate $\hat{R}_1, \ldots, \hat{R}_K$ for $T$.
3 Compute the centroids $\bar{X}_k = |\hat{R}_k|^{-1} \sum_{j \in \hat{R}_k} X_j$ and within-group sample covariance matrices $\hat{\Sigma}_k = |\hat{R}_k|^{-1} \sum_{j \in \hat{R}_k} (X_j - \bar{X}_k)(X_j - \bar{X}_k)^T$ for $k \in [K]$.
4 (Lift) For each $i \in [n] \setminus T$, assign $i \in \hat{G}_k$ if
$\log|\hat{\Sigma}_k| + \|\hat{\Sigma}_k^{-1/2}(X_i - \bar{X}_k)\|^2 < \log|\hat{\Sigma}_l| + \|\hat{\Sigma}_l^{-1/2}(X_i - \bar{X}_l)\|^2, \quad \forall l \neq k, l \in [K]$. And randomly assign $i$ to any $K$ clusters if such $k$ doesn't exist.
**Output:** A partition estimate $\hat{G}_1, \ldots, \hat{G}_K$ for $[n]$.

---

entry as 1, and 0 otherwise. The centers of clusters $\mu_1, \ldots, \mu_K$ are placed on vertices of a regular simplex, i.e., $\mu_k = \lambda\sqrt{1 + (1 + L)^{-1}}e_k, \ k \in [K]$. This ensures that for any $L$, $\Delta = \lambda, \ \forall \lambda$. From Figures 2 we can observe that the signal-to-noise ratio we defined is reasonable. On the other hand, the performance of SDP becomes worse as condition number of the group covariance matrices grows since the assumption of isotropy group covariance matrices for SDP is violated and same reason for $K$-means.

**Impact of dimension reduction.** Here we want to see the performance of iLA-SDP after dimension reduction. The covariance matrices of GMM are drawn independently following $\Sigma_k := U_k \Lambda_k U_k^T, \ \forall k \in [K]$. Here $U_k$ is a random orthogonal matrix, $\Lambda_k$ is a diagonal matrix with entries drawn from $\mathcal{Z} = 1 + \beta Z \cdot \mathbf{1}(Z > 0)$, where $Z$ is standard Gaussian distribution, $\beta > 0$ controls the condition number of $\Sigma_k$. Here we choose $n = 200, \ p = 20, \ K = 4, \ \beta = 5$. The covariance matrices are fixed once chosen and we perform Algorithm 1 on the dataset directly to get the results of iLA-SDP for each replicates. For dimension reduction, we follow the procedure of dimension reduction introduced in Algorithm 2 and Algorithm 3 in Appendix A.2 and get the transformed dataset $\tilde{X}$ with lower dimension. Then the results of pLA-SDP is obtained from running Algorithm 1 with HC as initialization on $\tilde{X}$. The results in Figure 6 shows that after reduction of dimension in our procedure, the performance of iLA-SDP becomes significantly better when the separation is large. This is because in our setting, the difference between centers $d_{(k,l)} := \mu_k - \mu_l$, is

sparse for all distinct pairs. And after performing the F-test on the covariates, the noisy terms get eliminated which results in better performance.

**Failure of EM vs SDP.** Recall the failure of EM for random initialization (Jin et al., 2016) in the special case that covariance matrices equal to identity matrix and it assumes equal weights. Both covariance matrices and weights are known. In this case, EM algorithm would be reduced to the version that the weights and the mean update interactively. Meanwhile, iLA-SDP would be reduced to SDP. The random initialization indicates that we pick any data point as initialization of the centers uniformly. Following the same setting from the construction of the pitfall, we choose one dimension GMM with three clusters such that the distance between two of the centers is much smaller than others. More concisely, we let $n = 300$, $K = 3$, $p = 1$, $\mu_1 = \gamma$, $\mu_2 = -\gamma$, $\mu_3 = 10 \cdot \gamma$. The results can be observed from the first plot in Figure 3 with 300 replicates, where we denote the reduced version of EM as mEM. From the figure we can observe that the reduced version of iLA-SDP, which is SDP, performs stable and achieves exact recovery when the separation is large. However, EM would fail for random initialization.

**Perturbation of initialization assignments.** To see how the performance of EM and iLA-SDP will change when perturbing the initialization, we set HC as initialization and proportion $\alpha$ ($\alpha \in [0, 1]$) of the initialization labels will be perturbed. The diagonal of the covariance matrices are placed at a simplex of $\mathbb{R}^p$ that are not identical to the corresponding centers. i.e. $\mu_k = \lambda \cdot e_k$, $\Sigma_k = L \cdot \mathrm{diag}(e_{k+1})$, $\forall l \in [K]$, where $e_{K+1} = e_1$. This guarantees the symmetry of the construction. We set $L = 10$, $p = 4$, $K = 4$ and the distance between centers $d = 8$. Each time we draw the $n = 200$ data from the GMM and run HC as initialization. Then we randomly assign $\alpha$ proportion of the labels from HC to any cluster uniformly. The results of the simulation for the second plot in Figure 3 are obtained through 300 total replicates, where we can observe that iLA-SDP is fairly stable with perturbation of initialization if the separation is large while EM can go worse as $\alpha$ approaches 1, i.e., all the labels are selected randomly. In other words, EM is more sensitive to initialization and iLA-SDP is more stable if the signal is strong.

**Empirical evidence for monotone increasing of objective function for iLA-SDP.** Here we provide examples based on previous experiment settings where we set the distance between centers $d = 1/3/5/10$. and try to see how the log-likelihood function of given data changes as the iteration proceeds. From Figure 7 in Appendix we can see that our algorithm guarantees that the log-likelihood function of given data increases over iteration empirically. What is more, by our construction we can show that the log-likelihood function will increase after each step for iLA-SDP theoretically.

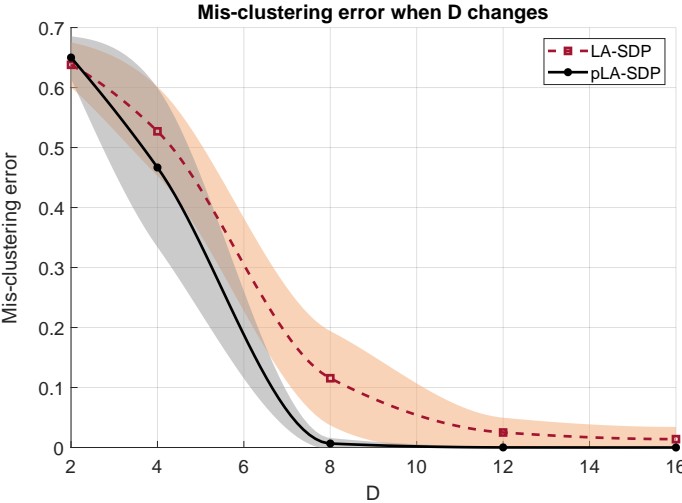

Figure 6: Mis-clustering error (with shaded error bars for the left plot) vs center distance $D$ for iLA-SDP before and after dimension reduction. pLA-SDP denotes the iLA-SDP after dimension reduction.

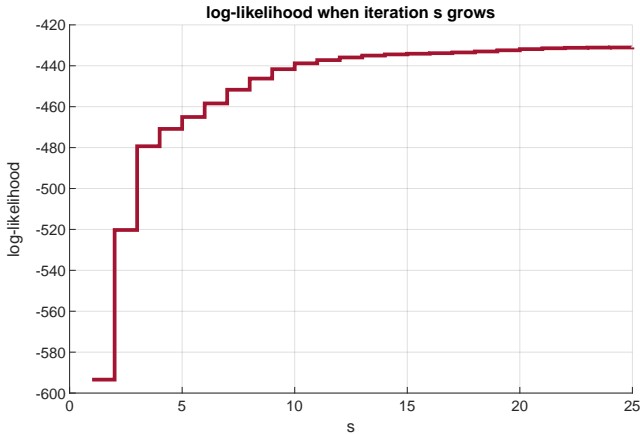

Figure 7: Log-likelihood (up to some constant) as iteration $s$ grows for iLA-SDP.

### A.4 PROOF OF THE THEOREMS AND PROPOSITIONS

In this section, we provide the proofs for the Proposition 1, Proposition 4 and a sketch proof of Theorem 2. The proof of the main theorem follows the track from the paper solving the exact recovery for original SDP (Chen & Yang, 2021b) and we will show the main differences in our proof.

First, we provide explicit expressions of some constants appearing in Theorem 2 below:

$$E_1 = \frac{4(1+2\delta)M^{5/2}}{(1-\beta)^2\eta^2}\left(M + \sqrt{M^2 + \frac{(1-\beta)^2}{(1+\delta)}\frac{p}{m\log n} + C_4 R_n}\right)$$

with

$$R_n = \frac{(1-\beta)^2}{(1+\delta)\log n}\left(\frac{\sqrt{p\log n}}{\underline{n}} + \frac{\log n}{\underline{n}}\right),$$

and

$$E_2 = \frac{C_5(M-1)^3 M^2}{(1-\beta)(1-\eta)}\left(\frac{p}{\log n} + 1\right) + \frac{C_6 K^2(1-\beta)}{\beta}$$

$$\cdot \min\left\{\frac{1}{\beta(M-1)^2}\frac{n}{m}\left(1 + \frac{\log p}{\log n}\right)\frac{p}{\log n}, \frac{(M-1)M^2}{\beta}\left(\sqrt{\frac{p^3}{\log n}} + \sqrt{p\log n}\right)\frac{n}{\sqrt{m}}\right\}. \tag{15}$$

### A.4.1 Proof of Proposition 1

*Proposition 1* (**SDP relaxation for $K$-means is a special case of LA-SDP**). Suppose $\Sigma_k = \sigma^2\mathrm{Id}_p$ for all $k \in [K]$. Let $\hat{Z}$ be the solution to (5) that achieves maximum $M_1$ and $\hat{Z}_k, k = 1, \ldots, K$, be the solution to (5) with maximum $M_2$. Then $M_1 = M_2$. And $\hat{Z} = \sum_{k=1}^{K}\hat{Z}_k$, if $\hat{Z}$ is unique in (5).

*Proof of Proposition 1* If $\Sigma_k = \sigma^2\mathrm{Id}_p$, $\forall k \in [K]$. Then from (7) we have

$$A_k \equiv \frac{1}{2}\left[\mathrm{diag}(X^T X)\mathbf{1}_n^T + \mathbf{1}_n\mathrm{diag}(X^T X)^T\right] + X^T X, \ \forall k \in [K].$$

This implies that (8) can be written as

$$\hat{Z}_1, \ldots, \hat{Z}_K = \arg\max_{Z_1, \ldots Z_K \in \mathbb{R}^{n\times n}}\left\langle X^T X, \left(\sum_{k=1}^{K} Z_k\right)\right\rangle \tag{16}$$

$$\text{subject to } Z_k \succeq 0, \ \mathrm{tr}\left(\sum_{k=1}^{K} Z_k\right) = K, \ \left(\sum_{k=1}^{K} Z_k\right)\mathbf{1}_n = \mathbf{1}_n, \ Z_k \geqslant 0, \ \forall\, k \in [K],$$

Since $\left\langle\mathrm{diag}(X^T X)\mathbf{1}_n^T, \left(\sum_{k=1}^{K} Z_k\right)\right\rangle = \mathrm{tr}(X^T X)$, which is a constant in the optimization problem (16). Now suppose $\hat{Z}$ is a solution to (5) that achieves maximum $M_1$ and $\hat{Z}_k, k = 1, \ldots, K$, is the solution to (16) that achieves maximum $M_2$, then we have

$$\left\langle X^T X, \left(\sum_{k=1}^{K} Z_k\right)\right\rangle \leq M1,$$

$$\left\langle X^T X, \left(\sum_{k=1}^{K} \tilde{Z}_k\right)\right\rangle \leq M2,$$

where $\tilde{Z}_1 := \hat{Z}$, $\tilde{Z}_2 = \cdots = \tilde{Z}_K = 0$. In other words, $M_1 = M_2$, which finishes the proof. If $\hat{Z}$ is unique in (5), then we have $\hat{Z} = \sum_{k=1}^{K}\hat{Z}_k$ since both of them achieve the maximum in (5). $\blacksquare$

### A.4.2 Proof of Proposition 4

*Proposition 4* (**iLA-SDP is a soft clustering method**). If $\mathrm{rank}(Z_k) = 1$, then there exists weights $(w_{k,1}, \ldots, w_{k,n})$ such that $\hat{\Sigma}_k$ in Lemma 3 can be written as

$$\hat{\Sigma}_k := \frac{1}{n_k}\sum_{i=1}^{n} w_{k,i}(X_i - \hat{\mu}_k)(X_i - \hat{\mu}_k)^\top \quad\text{with}\quad \hat{\mu}_k := \frac{1}{n_k}\sum_{i=1}^{n} w_{k,i}X_i, \tag{17}$$

where $n_k = \sum_{i=1}^{n} w_{k,i}$.

*Proof of Proposition 4* If $Z_k$ is rank 1, then there exists $a \in \mathbb{R}^n$ such that $Z_k = aa^T$. Let $w_k := a^T \mathbf{1} \cdot a$, then we have

$$Z_k = \frac{w_k w_k^T}{w_k^T \mathbf{1}},$$

i.e., $Z_{k,ij} = \frac{w_{k,i} w_{k,j}}{\sum_{i=1}^n w_{k,i}}$. Finally, by plugging in the expression of $Z_{k,ij}$ with $w_{k,i}$ we can get the target expression for $\hat{\Sigma}_k$. ∎

### A.4.3 Sketch proof of Theorem 2

*Theorem 2* (**Exact recovery for LA-SDP**). Suppose there exist constants $\delta > 0$ and $\beta \in (0,1)$ such that

$$\log n \geq \max \left\{ \frac{(1-\beta)^2}{\beta^2}, \frac{(1-\beta)(1-\eta)K^2}{\beta^2 \max\{(M-1)^2, 1\}} \right\} \frac{C_1 n}{m}, \quad \delta \leq \frac{\beta^2}{(1-\beta)^2} \frac{C_2 M^{1/2}}{K}, \quad m \geq \frac{4(1+\delta)^2}{\delta^2}.$$

If

$$\Delta^2 \geq (E_1 + E_2) \log n, \quad \text{and} \quad \min_{k \neq l} D_{(k,l)} \geq C_3(1 + \log n/p + p/n), \tag{18}$$

where

$$E_1 = \frac{4(1+2\delta)M^{5/2}}{(1-\beta)^2 \eta^2} \left( M + \sqrt{M^2 + \frac{(1-\beta)^2}{(1+\delta)} \frac{p}{m \log n} + C_4 R_n} \right)$$

with

$$R_n = \frac{(1-\beta)^2}{(1+\delta)\log n} \left( \frac{\sqrt{p \log n}}{\underline{n}} + \frac{\log n}{\underline{n}} \right),$$

and

$$E_2 = \frac{C_5(M-1)^3 M^2}{(1-\beta)(1-\eta)} \left( \frac{p}{\log n} + 1 \right) + \frac{C_6 K^2 (1-\beta)}{\beta}$$
$$\cdot \min \left\{ \frac{1}{\beta(M-1)^2} \frac{n}{m} \left( 1 + \frac{\log p}{\log n} \right) \frac{p}{\log n}, \frac{(M-1)M^2}{\beta} \left( \sqrt{\frac{p^3}{\log n}} + \sqrt{p \log n} \right) \frac{n}{\sqrt{m}} \right\}; \tag{19}$$

then the LA-SDP achieves exact recovery, or $\hat{Z} = Z^*$, with probability at least $1 - C_7 K^3 n^{-\delta}$ for some universal constants $C_1, \ldots, C_7$.

*Sketch of the proof.* Recall that we let $G_1^*, \ldots, G_K^*$ be the true partition of the index set $[n] := \{1, \ldots, n\}$ such that if $i \in G_k^*$, then

$$X_i = \mu_k + \epsilon_i, \tag{20}$$

where $\mu_k \in \mathbb{R}^p$ is the true center of the $k$-th cluster $G_k^*$ ($G_k$ for simplicity) and $\epsilon_i$ is an i.i.d. random Gaussian noise $N(0, \Sigma_k)$. First we can write down the dual problem:

$$\min_{\substack{\lambda \in \mathbb{R}, \alpha \in \mathbb{R}^n, \\ B_k \in \mathbb{R}^{n \times n}}} \lambda K + \alpha^T \mathbf{1}_n, \quad \text{subject to } B_k \geq 0, \ \lambda \mathrm{Id}_n + \frac{1}{2}(\alpha \mathbf{1}_n^T + \mathbf{1}_n \alpha^T) - A_k - B_k \succeq 0, \ \forall k \in [K].$$

Denote $Z_k^* := \frac{1}{|G_k|} \mathbf{1}_{G_k} \mathbf{1}_{G_k}^T$, $\forall k \in [K]$ then it can be shown that the sufficient conditions for the solution of SDP to be $Z_k = Z_k^*$, $\forall k \in [K]$ are

$$B_k \geq 0; \tag{C1}$$

$$W_k := \lambda \mathrm{Id}_n + \frac{1}{2}(\alpha \mathbf{1}_n^T + \mathbf{1}_n \alpha^T) - A_k - B_k \succeq 0; \tag{C2}$$

$$\mathrm{tr}(W_k Z_k^*) = 0; \tag{C3}$$

$$\mathrm{tr}(B_k Z_k^*) = 0. \tag{C4}$$

It can be verified that if we can find symmetric $B_k$ such that

$$B_{k,G_k G_k} = 0; $$

$$
[B_{k,G_l G_k} \mathbf{1}_{G_k}]_i = -\frac{n_k + n_l}{2n_l} \cdot \lambda
$$
$$
+ \frac{n_k}{2} [(\|\Sigma_k^{-1/2}(\bar{X}_k - X_i)\|^2 + \log |\Sigma_k|) - (\|\Sigma_l^{-1/2}(\bar{X}_l - X_i)\|^2 + \log |\Sigma_l|)];
$$
$$
[B_{k,G_l G_l} \mathbf{1}_{G_l}]_j = [A_{l,G_l G_l} \mathbf{1}_{G_l}]_j - [A_{k,G_l G_l} \mathbf{1}_{G_l}]_j;
$$
$$
[B_{k,G_{l'} G_l} \mathbf{1}_{G_l}]_j = [B_{l,G_{l'} G_l} \mathbf{1}_{G_l}]_j + [A_{l,G_{l'} G_l} \mathbf{1}_{G_l}]_j - [A_{k,G_{l'} G_l} \mathbf{1}_{G_l}]_j,
$$

for any triple pairs $(k, l, l')$ that are mutually distinct and $i \in G_k$, $j \in G_l$. Then (C3) and (C4) hold. In fact, the target matrices can be defined through

$$
B_{k,G_{l'} G_l}^{\#} := \frac{B_{k,G_{l'} G_l} \mathbf{1}_{G_l} \mathbf{1}_{G_{l'}}^T B_{k,G_{l'} G_l}}{\mathbf{1}_{G_{l'}}^T B_{k,G_{l'} G_l} \mathbf{1}_{G_l}}, \tag{21}
$$

for any $k \in [K]$, $(l', l) \neq (k, k)$. Furthermore, the construction of $B_k$ shows that $B_k \mathbf{1}_{G_l} = 0$, $\forall (k, l)$ pairs.

The following two lemma gives the sufficient conditions for (C1).

*Lemma* 6 (**Separation bound on the covariance matrices**). Let $\lambda_1, \ldots, \lambda_p$ correspond to the eigenvalues of $(\Sigma_l^{1/2} \Sigma_k^{-1} \Sigma_l^{1/2} - \mathrm{Id}_p)$ and define $D_{(k,l)} := \frac{\sum_{i=1}^p (\lambda_i - \log(1 + \lambda_i))}{p \max_i |\lambda_i|}$. If there exists constant $C$ such that

$$
\min_{k \neq l} D_{(k,l)} \geq C(1 + \log n/p + p/n),
$$

then

$$
\mathbb{P}\Big([A_{l,G_l G_l} \mathbf{1}_{G_l}]_j - [A_{k,G_l G_l} \mathbf{1}_{G_l}]_j \geq 0, \text{ for all } (k, l) \in [K]^2 \text{ and } j \in G_l\Big) \geq 1 - CK^2/n.
$$

*Lemma* 7 (**Separation bound on the centers**). Let $\delta > 0$, $\beta \in (0, 1)$, $\eta \in (0, 1)$. If we have

$$
\Delta^2 \geq \frac{4(1 + \delta)M^2}{(1 - \beta)^2 \eta^2} \left[ M^{3/2} + \sqrt{M^3 + \frac{(1 - \beta)^2 M}{(1 + \delta)} \frac{p + 2\sqrt{p \log(nK)} + 4 \log(nK)}{m \log n}} \right] \log n,
$$

and

$$
\Delta^2 \geq \frac{M^2 (M - 1)^2}{(1 - \beta)^2 (1 - \eta)^2} \cdot
$$
$$
\left(1 + \frac{2(1 - \beta)(1 - \eta)}{M} [3 \log M + 4M(M - 1)(p + 2\sqrt{p \log(nK)} + 4 \log(nK))]\right),
$$

then

$$
\mathbb{P}\Big(\|\Sigma_l^{-1/2}(\bar{X}_l - X_j)\|^2 + \log |\Sigma_l|) - (\|\Sigma_{l'}^{-1/2}(\bar{X}_{l'} - X_j)\|^2 + \log |\Sigma_{l'}|)
$$
$$
- \frac{2}{n_l} |[A_{l,G_{l'} G_l} \mathbf{1}_{G_l}]_j - [A_{k,G_{l'} G_l} \mathbf{1}_{G_l}]_j| \geq \frac{\beta}{M} \|\Sigma_l^{-1/2}(\mu_l - \mu_{l'})\|^2 + (n_l^{-1} + n_{l'}^{-1})p - r_{k,l,l'},
$$

for all triple $(k, l, l') \in [K]^3$ with $(k, l, l') \neq (k, k, k)$ and $j \in G_{l'}\Big)
$$
$$
\leq \frac{CK^3}{n^\delta},
$$

where

$$
r_{k,l,l'} = 4\sqrt{\frac{\log(nK)}{n_l}} \|\Sigma_l^{-1/2}(\mu_l - \mu_{l'})\| + 2(n_l^{-1} + n_{l'}^{-1})\sqrt{2p \log(nK)} + 4n_{l'}^{-1} \log(nK).
$$

for some large constant $C$.

The proof of Lemma 7 follows the similar steps from the original paper (Chen & Yang, 2021b). The two lemmas imply that (C1) can hold with high probability if the separation condition in the assumption holds. The remaining part is to verify the (C2).

Denote $\Gamma = \text{span}\{\mathbf{1}_{G_k} : k \in [K]\}^\perp$ be the othogonal complement of the linear space spanned by $\mathbf{1}_{G_k}$, $k \in [K]$. Note that $W_k \mathbf{1}_{G_l} = 0$, $\forall (k, l) \in [K]^2$, we only need to check for $v \in \Gamma$,

$$v^T W_k v \geq 0, \ \forall k \in [K].$$

Note that $v^T \mathbf{1}_{G_k} = 0$, we have

$$v^T W_k v = \lambda \|v\|^2 - S_k(v) - T_k(v),$$

where $S_k(v) := v^T A_k v = v^T X^T \Sigma_k^{-1} X v$, and $T_k(v) = v^T B v$. By concentration bound we can get

$$\mathbb{P}(S_k(v) \leq MK(\sqrt{n} + \sqrt{p} + \sqrt{2\log n}), \text{ for all } k \in [K]) \geq 1 - \frac{K}{n}.$$

For $T_k(v)$, first we define

$$V_{k,ll'}^{(1)} := \langle \Sigma_{l'}^{1/2} \Sigma_l^{-1}(\mu_{l'} - \mu_l), \sum_{j \in G_{l'}} v_j \epsilon_j \rangle;$$

$$V_{k,ll'}^{(2)} := \langle \bar{\epsilon}_{l'} - \Sigma_{l'}^{1/2} \Sigma_l^{-1/2} \bar{\epsilon}_l, \sum_{j \in G_{l'}} v_j \epsilon_j \rangle;$$

$$V_{k,ll'}^{(3)} := \frac{1}{2} \sum_{j \in G_{l'}} \epsilon_j^T \Sigma_{l'}^{1/2}(\Sigma_l^{-1} - \Sigma_{l'}^{-1})\Sigma_{l'}^{1/2} \epsilon_j v_j;$$

$$V_{k,ll'}^{(4)} := \frac{1}{n_l} \sum_{j \in G_{l'}} ([A_{l,G_{l'}G_l}\mathbf{1}_{G_l}]_j - [A_{k,G_{l'}G_l}\mathbf{1}_{G_l}]_j) v_j \cdot \mathbf{1}(l \neq l').$$

Then we can write $T_k(v)$ as

$$T_k(v) := \sum_{l \neq l'} \frac{n_l n_{l'}}{\mathbf{1}_n^T B_k \mathbf{1}_n}(T_{k,ll'}^{(1)} + T_{k,ll'}^{(2)} + T_{k,ll'}^{(3)} + T_{k,ll'}^{(4)} + T_{k,ll'}^{(5)}),$$

where

$$T_{k,ll'}^{(1)} := V_{k,ll'}^{(1)} \cdot V_{k,l'l}^{(1)};$$

$$T_{k,ll'}^{(2)} := V_{k,ll'}^{(2)} \cdot V_{k,l'l}^{(2)};$$

$$T_{k,ll'}^{(3)} := V_{k,ll'}^{(1)} \cdot V_{k,l'l}^{(2)} + V_{k,ll'}^{(2)} \cdot V_{k,l'l}^{(1)};$$

$$T_{k,ll'}^{(4)} := (V_{k,ll'}^{(3)} + V_{k,ll'}^{(4)}) \cdot (V_{k,l'l}^{(1)} + V_{k,l'l}^{(2)}) + (V_{k,ll'}^{(1)} + V_{k,ll'}^{(2)}) \cdot (V_{k,l'l}^{(3)} + V_{k,l'l}^{(4)});$$

$$T_{k,ll'}^{(5)} := (V_{k,ll'}^{(3)} + V_{k,ll'}^{(4)}) \cdot (V_{k,l'l}^{(3)} + V_{k,l'l}^{(4)}).$$

Now we choose $\lambda = p + \frac{\beta}{4M}m\Delta^2$, which implies that

$$\mathbf{1}_n^T B_k \mathbf{1}_n \geq \frac{n_l n_{l'}}{8} \frac{\beta}{M} \max\{\|\Sigma_{l'}^{-1/2}(\mu_l - \mu_{l'})\|^2, \|\Sigma_l^{-1/2}(\mu_l - \mu_{l'})\|^2\}.$$

From concentration bounds for Gaussians we have for all triple $(k, l, l') \in [K]^3$ such that $(k, l, l') \neq (k, k, k)$,

$$\left|\sum_{l \neq l'} \frac{n_l n_{l'}}{\mathbf{1}_n^T B_k \mathbf{1}_n} T_{k,ll'}^{(1)}\right| \leq \frac{CM^2}{\beta} \cdot (n + \sqrt{2nK\log n} + 2K\log n)\|v\|^2;$$

$$\left|\sum_{l \neq l'} \frac{n_l n_{l'}}{\mathbf{1}_n^T B_k \mathbf{1}_n} T_{k,ll'}^{(2)}\right| \leq \frac{CM^3}{1-\beta} \cdot (\delta\sqrt{mp\log n} + \sqrt{mp\log^7 n}/\underline{n})\|v\|^2;$$

$$\left|\sum_{l \neq l'} \frac{n_l n_{l'}}{\mathbf{1}_n^T B_k \mathbf{1}_n} T_{k,ll'}^{(5)}\right| \leq \frac{CK^2}{\beta} \cdot (\frac{1-\beta}{(M-1)^2 M}(p + pM\log p/\log n) + M(M-1))n\|v\|^2,$$

Or

$$\left|\sum_{l \neq l'} \frac{n_l n_{l'}}{\mathbf{1}_n^T B_k \mathbf{1}_n} T_{k,ll'}^{(5)}\right| \leq \frac{CK^2 M(1-\beta)(M-1)}{\beta} \cdot (\sqrt{\frac{p^3 m}{\log n}} + \sqrt{pm\log n})n\|v\|^2,$$

with probability $\geq 1 - CK^3/n^\delta$ for some constant $C$.

Note that by assumption we have $\Delta^2 \geq \frac{C(M-1)^3 M^2}{(1-\beta)(1-\eta)}(p + \log n) + \frac{CM^3}{(1-\beta)}\sqrt{(1+\delta)p \log n/m}$ and the fact that the remaining terms of $T_{k,ll'}$ can be bounded by the above inequalities up to multiplied by some constant, we can directly verify that (C2) is true under our assumptions. ∎

*Lemma 6* (**Separation bound on the covariance matrices**). Let $\lambda_1, \ldots, \lambda_p$ correspond to the eigenvalues of $(\Sigma_l^{1/2}\Sigma_k^{-1}\Sigma_l^{1/2} - \mathrm{Id}_p)$ and define $D_{(k,l)} := \frac{\sum_{i=1}^p (\lambda_i - \log(1+\lambda_i))}{p \max_i |\lambda_i|}$. If there exists constant $C$ such that

$$\min_{k \neq l} D_{(k,l)} \geq C(1 + \log n/p + p/n),$$

then

$$\mathbb{P}\Big([A_{l,G_l G_l}\mathbf{1}_{G_l}]_j - [A_{k,G_l G_l}\mathbf{1}_{G_l}]_j \geq 0, \text{ for all } (k,l) \in [K]^2 \text{ and } j \in G_l\Big) \geq 1 - CK^2/n.$$

*Sketch of the proof.* Let $T := [A_{l,G_l G_l}\mathbf{1}_{G_l}]_j - [A_{k,G_l G_l}\mathbf{1}_{G_l}]_j$, $B := \Sigma_l^{1/2}\Sigma_k^{-1}\Sigma_l^{1/2} - \mathrm{Id}_p$ then by definition we have

$$
\begin{aligned}
T = &-\sum_{i=1}^p \log(\lambda_i + 1) + \sum_{i=1}^p \lambda_i \\
&+ \frac{1}{2}\langle B, \epsilon_j \epsilon_j^T - \mathrm{Id}_p \rangle \\
&- \frac{1}{2}\langle B, \frac{1}{n_l}\sum_{t \in G_l}\epsilon_t \epsilon_j^T + \epsilon_j\Big(\frac{1}{n_l}\sum_{t \in G_l}\epsilon_t\Big)^T \rangle \\
&+ \frac{1}{2}\langle B, \frac{1}{n_l}\sum_{t \in G_l}\epsilon_t \epsilon_t^T - \mathrm{Id}_p \rangle,
\end{aligned}
$$

where the last three terms can be bounded by concentration bounds for Gaussians. ∎

