# OpenReview forum: "Likelihood adjusted semidefinite programs for clustering heterogeneous data"
_ICLR.cc/2023/Conference — Submitted to ICLR 2023_

### Official Review · Reviewer_FBAZ · 2022-10-21

**Confidence:** 2
**Correctness:** 4
**Technical Novelty And Significance:** 2
**Empirical Novelty And Significance:** 2
**Recommendation:** 3

**Clarity, Quality, Novelty And Reproducibility:**

The paper reads well and clear, however, novelty is limited and so is reproducibility as the authors have not shared code for the experiments.

**Strength And Weaknesses:**

Strength
======

- The paper is well written, except for the references/citation style. It's really hard to find the papers that you cited in the main manuscript in the References section as citations are done with last name, e.g., (Balakrishnan et al 2017) but in the Reference section they start with the author's first name, e.g., "Sivaraman Balakrishnan, Martin J. Wainwright..."
- The idea of optimizing cluster labels, while not novel (see, e.g., "Discrete Optimal Graph Clustering" by
Yudong Han et al), is interesting.

Weaknesses
=========

- Experiments with real data are very limited and do not consider classical methods for clustering such as Spectral Clustering.

Minor comments
=============

- This sentence: "Another popular clustering method is the classic expectation-maximization (EM) algorithm" does not make sense. EM is not a clustering method. EM is an iterative algorithm for finding local minima of (generally) maximum likelihood problems.

**Summary Of The Paper:**

This paper extends an SDP approach for Gaussian mixture models clustering by introducing cluster labels as model parameters.

**Summary Of The Review:**

The idea of the paper is interesting, but the lack of experimental results with popular and successful clustering methods diminishes the paper's overall contribution to the community.

---

> ### Author Response · Authors · 2022-11-18
> **We thank you for your comments on our work and appreciate all your precious advice.**
>
> (__On the experiments__) For the experiments on real datasets, we have uploaded side-by-side boxplots for the difference of mis-clustering rate in the revision in place of the former plots to make the comparison clear; we also include the spectral clustering method as another competitor. The codes for the main algorithm iLA-SDP will be provided as well.
> \
> \
> (__On the EM algorithm__) Thanks for pointing it out. We will change our phrase to ``Another popular model-based clustering method is the Gaussian mixture model where component parameters are estimated via the classic expectation-maximization (EM) algorithm." A well-known R package _mclust_ implemented the EM algorithm in the setting of GMM clustering. One may refer to _mclust 5: Clustering, Classification and Density Estimation Using Gaussian Finite Mixture Models by Luca Scrucca, Michael Fop, T. Brendan Murphy, and Adrian E. Raftery_ for more details.

---

### Official Review · Reviewer_BdFu · 2022-10-24

**Confidence:** 3
**Correctness:** 3
**Technical Novelty And Significance:** 3
**Empirical Novelty And Significance:** 2
**Recommendation:** 5

**Clarity, Quality, Novelty And Reproducibility:**

The paper is generally well written.  The extension of the SDP-relaxation to the model-based clustering case seems novel.  Scholarship is a bit on the weaker side, and important references are missing.    The paper is mostly of theoretical interest, as the method seems very expensive compared to competition.  Experiments are on the weaker side, and it's unclear if the competition k-means, EM e.t.c. were given all the standard options (e.g. k-means++ init,  or random restarts) to reach their potential -- given that they're significantly cheaper.

**Strength And Weaknesses:**

Strengths:  The paper proposes an interesting SDP relaxation for model-based clustering with known covariance matrices, and extends the information theoretic analysis from the isotropic case.  In the usual case where within-cluster covariances are now known,  the paper proposes an iterative approach which alternates between covariance estimation with known clusters, and SDP-relaxad covariance estimation with known covariances.  The approach is limited to small data (timing and computational complexity is not discussed in the paper, but SDP is very expensive and the largest simulation has 1000 data-points) -- but perhaps there could be critical small-data problems where any gains in clustering quality could be worth the computational cost.  The appendix mentions that it may be possible to extend the method to large data (larger dimensions, and more samples) -- but it's unclear how valuable/practical that is -- and it uses standard ideas like sub-sampling.

Weaknesses:
1) Computational complexity or timing is not discussed in the paper (which should definitely be addressed) -- and it's likely very high.  The authors mention SDP solvers to be O(N^3.5), and the relevant dimension N seems to be n*K  where n is data-points and K is the number of clusters. So it's useful at best for very small scale problems. When covariances are unknown -- the method is iterative, solving many such SDPs, and no information is provided of how many iterations are typically needed until progress slows down. It's worth reporting the timing of the proposed method and contrast with standard k-mean + k-means++ / EM -- to know the price to pay for the clustering accuracy improvements.

2) In the common case of unknown covariances -- the method appears more akin to EM, which the authors aim to avoid in the first place.  (There's discussion of why it still may be better than EM -- but it's not very convincing -- definitely the elegance of one-shot SDP is lost). Convergence or rates are no longer guaranteed. Also in practice the method uses a lot of other clustering methods (sometimes HC is used for initialization, and spectral-clustering and k-means are used for post-processing non-integral partitioning solutions) -- so it seems quite sensitive to initialization.  So while it's discussed as a clean one-step SDP relaxation -- in practice it's quite complex and uses various heuristics.

3) Experiments do show an improvement -- but they are on the weaker side, and important details are not provided.  Did you use k-means++ initialization for k-means, and something like that for EM?  What is the computational cost of your method w.r.t. alternatives?  If k-means runs say 1000 times faster -- you could probably use this budget for random restarts, some more accurate heuristic initialization schemes -- how would it compare then?  Other common clustering methods (e.g. spectral clustering are not evaluated), even though they're used in post-processing of SDP solutions.  In figure 4 -- the bulk of the box-plot for LA-SDP is in-line with other methods,  but it has a small set of outliers which do better -- this seems an artifact of this specific problem, and perhaps better initialization for k-means could find these configurations. Ideally more than 2 datasets would be used.  Mis-clustering error is not defined in the body of the paper.  Also some discussion in the short experimental section is confusing. You subsample clusters to keep them 'balanced' -- but there's a discussion of 'highly unbalanced'...
3-a) "Perturbation percentage of initaliation" is not defined.
3-b) Also why do you need graphical Lasso when you have 4 attributes?  Plain sample covariance should work very well even with relatively few samples that you have.

4) Insufficient scholarship.  It's unclear where exactly the SDP k-means method was proposed -- the main citation is from 2021, but it seems to have a much longer history, including other earlier paper analyzing its performance, and many relevant references are missing. If there are many qualitatively different SDP relaxations -- then they should be mentioned and contrasted, otherwise key papers introducing and analyzing the method should also be mentioned.  Or does your paper propose a new isotropic SDP?
* Iguchi, Takayuki, et al. "On the tightness of an SDP relaxation of k-means.", 2015.
* Moses Charikar, Rachel Ward, Awasthi, Pranjal, et al. "Relax, no need to round: Integrality of clustering formulations." Proc. Conference on Innovations in Theoretical Computer Science. 2015.

5) The paper claims that profiling out the cluster-centers (i.e. not keeping them explicit) is a major advantage.  It's a key step in the SDP formulation, but once you have a (soft) cluster assignment -- cluster centers are trivially computed, why is it so important?

 6) Term "profiled-out" commonly used in the paper is never defined -- setting nuisance parameters to their maximum-likelihood values (rather than being integrated out).













**Summary Of The Paper:**

The paper extends an SDP relaxation for k-means clustering (Gaussian mixture modeling) from isotropic clusters (with identical scaled identity covariances) to non-isotropic clusters, where each cluster can have its own covariance matrix.  When these cluster covariances are unknown (which is most often the case) the paper proposes an iterative EM-like algorithm which alternates between covariance estimation with known cluster-labels, and SDP-relaxed model-based clustering with known covariances.  Limited simulations show improvements w.r.t. k-means, EM and hierarchical clustering.

**Summary Of The Review:**

Interesting extension of SDP clustering and its analysis to the model-based clustering setting with non-isotropic covariances.  The proposal is likely only of theoretical interest, as the computational cost is very high.  Also in the practical setting of unknown covariances the method looses its convex-relaxation elegance, and starts looking more like EM that it tries to replace, requiring various heuristics.  Scholarship, and experiments need improvement.

---

> ### Author Response · Authors · 2022-11-18
> **We thank you for your comments on our work and appreciate all your useful feedback.**
>
> (__On the computational complexity and spectral clustering__)  Please refer to Appendix A.1 in the revision for the comparison of time cost. Spectral clustering has been added in Section 4.
> \
> \
> (__On the iLA-SDP__) \
> _For the initialization issues_. As we know, EM is sensitive to the initialization assignments even when the separation is large as we shown in the paper. However, iLA-SDP is more stable to initialization when the separation is large. Since iLA-SDP is affected by initialization assignments only through covariance matrices, while EM is affected by initialization assignments through both cluster means and covariance matrices.
> \
> _Regarding the comparison with the original SDP_: even it has a clean one-step formation, its accuracy is not good when the covariance matrices of clusters are significantly different. This is due to the fact that SDP assumes the covariance matrices to be isometric. To make SDP more general and adaptive to different situations, we need to treat the covariance matrices as parameters. Our proposed method is based on alternating maximizing algorithm and it is plausible and natural to take initialization into consideration.  Currently, we can prove the convergence of iLA-SDP (From Lemma 3 and the construction of oracle LA-SDP we know the objective function is monotonically increasing across the iterations). We also have extensive numerical evidences demonstrating the convergence, which is shown as Figure 6 in the Appendix. However, deriving the convergence rate is quite challenging (as EM algorithm) and it will be our future goal to address it.
> \
> \
> (__On the experiments__) For the experiments for the real datasets, we have uploaded side-by-side boxplots for the difference of mis-clustering rate in the revision in place of the former plots to make the comparison clear.\
> (1) We use HC as initialization for EM, K-means and iLA-SDP as we mentioned in Section 4.\
> (2) If we use random initialization for $K$-means instead of HC, the comparisons to other methods (with HC as initialization) are not clear as we are focusing on comparison here. \
> (3) The outliers cannot be addressed by the initialization for $K$-means. As you can imagine, $K$-means put isotropic assumption on the covariance matrices, where the results aligned with SDP as expected. That is why EM/iLA-SDP out perform other methods for these subsets of the data.\
> (4) Mis-clustering error stands for the error rate. The proportion of data that have been assigned to the wrong clusters.\
> (5) The "highly unbalanced data" mentioned in Section 4 referred to the clustering results from HC (as initialization). The data for each replicate are balanced.\
> (6) Perturbation percentage of the initialization is the percentage of the labels for the initialization that have been randomly reassigned to any cluster uniformly.\
> (7) For the case when the assignments of HC initialization are not highly balanced, we consider the covariance matrices to be diagonal to make smooth connection between EM/iLA-SDP and original SDP/K-means methods. When the assignments of HC initialization are highly balanced, which indicates that the covariance matrices are more important, we use iLA-SDP/EM directly. Graphic LASSO here is merely served as a way to diagonalize the covariance matrices during iteration for iLA-SDP.
> \
> \
> (__On the scholarship__) The earlier references for SDP has been cited and discussed in the third paragraph in Introduction.
> \
> \
> (__On profiling out the cluster-centers__) As you mentioned, the cluster centers are fully determined given the assignments, which indicates that the centers are highly correlated to the assignments. This would result in problematic performance due to extremely slow convergence and high sensitivity to initialization for iterative algorithms like EM where the updates for the assignments (E-step) and centroids (M-step) are separated. In our proposed iLA-SDP, by profiling out the correlated centroids, one can think of updating the membership matrices is automatically grouped with the centroids update via profiling and thus we obtain better performance in terms of robustness to initialization and stability for high-dimensional data. This is a major benefit of the iLA-SDP over the EM algorithm.
> \
> \
> (__On Profiling out and Marginalizing__) Thanks for reminder. In fact, "profiled-out" is a standard terminology in statistics meaning that by first expressing the nuisance parameters as functions of target parameters via maximum likelihood with fixed target parameters and then maximize the likelihood in the remaining target parameters. In the clustering setting under the Gaussian mixture model, profiling out the centroids turns out to be independent of target parameters $(\Sigma_k, Z_k)$ in LA-SDP, which optimize profiled likelihood for these target parameters. Thus, profiled-out is different from integrated-out where the latter means subset parameters are treated as random variables and will be marginalized out.

---

### Official Review · Reviewer_cBBj · 2022-10-25

**Confidence:** 3
**Correctness:** 3
**Technical Novelty And Significance:** 2
**Empirical Novelty And Significance:** 2
**Recommendation:** 5

**Clarity, Quality, Novelty And Reproducibility:**

The writing is clear and easy to follow for most parts. Since this is mainly a theoretic work, it will be great if the authors could highlight their technical contributions, which seem unclear / lacking in the current shape.

**Strength And Weaknesses:**

- The MIP - SDP relaxation is very standard (see e.g. [1]).

- The authors highlight multiple times in the paper, that the technical novelty comes from extending the isotropic Gaussian covariances, to a more general setting where there is no such uniform and identical assumption. To me this sounds technically incremental, because the whole SDP framework stays the same, and it only requires some changes about the concentration inequalities. But I think this could also due to writing. Could the authors elaborate on this? Specifically, what are the technical novelties in the oracle LA-SDP analysis?

- Do the authors know whether the SDP relaxation is tight or not? More specifically, if the sample complexity condition in Theorem 2 is violated, are we immediately in a regime such that exact recovery is guaranteed to fail with large probability (information-theoretic lower bound)? Or we have something statistically possible but cannot be solved efficiently by any algorithm (computational lower bound)?

- Theorem 2: Can the authors provide some example settings, in which the sample complexity condition is fulfilled or not fulfilled? In other words, how should we interpret the sample complexity requirement?

- Is there any convergence rate guarantee on the iterative algorithm iLA-SDP, or is it purely heuristical? It is known that solving a single SDP is very time consuming (O(n^6) if I remember correctly), not to mention solving SDPs iteratively.

- Experiments: Do these datasets follow the rate condition in Theorem 2, or they only serve to show the effectiveness of the proposed algorithm in practice?



References:
- [1] Amini, Arash A., and Elizaveta Levina. "On semidefinite relaxations for the block model." The Annals of Statistics 46.1 (2018): 149-179.

**Summary Of The Paper:**

The paper studies the problem of clustering samples generated from multiple Gaussian mixtures with possibly non-identical and non-isotropic covariance matrices. The authors start with a mixed integer program (MIP) take the standard semidefinite program (SDP) relaxation approach. They first consider an oracle model, in which the covariances are known. For this oracle model the paper provides the sample complexity bound for exact recovery using SDP. Next the authors propose an iterative algorithm, which updates the estimated covariances and solves the new SDPs iteratively. Some real-world experiments are provided as well.

**Summary Of The Review:**

In the rebuttal it will be great if the authors could highlight their technical contributions, which seem unclear / lacking in the current shape.

---

> ### Author Response · Authors · 2022-11-18
> **We thank you for your comments on our work and appreciate all your helpful feedback.**
>
> (__On the technical novelties in the oracle LA-SDP analysis__) \
> (1) The formation of oracle LA-SDP (see formula (8) in Section 3.1) for clustering heterogeneous data is novel as far as we know. \
> (2) The dual construction in the proof is different from existing works. The proof of the zero dual gap is the core part for proving exact recovery for SDP. The dual construction boils down the proof to finding appropriate constructions for $K$ matrices $B_k,k\in[K]$ meeting a few constraints. The current construction requires more efforts (see equation (21) in Appendix A.4.3). Concretely, previous we only need to concern about one matrix $B,$ which consists of $K^2$ blocks indexed by $(k,l)\in[K]^2$. Now we need to consider $K$ matrices $B_k$ (see (C1) in Appendix A.3.1), which consists of $K\times K^2$ total blocks indexed by $(k,l,l')\in[K]^3.$ For different $k$, the blocks $B_{k,(G_l,G_{l'})}$ are different and need to be carefully addressed during construction. \
> (3) Recall the definition of $D_{k,l}$ (in the paragraph right before Theorem 2), it measures the pairwise-distance among covariance matrices of different clusters. In the existing SDP formulation for homogeneous data, we need $D_{k,l}=0$ so the $B_k$ collapses to $B$. Now stronger conditions must be put on $D_{k,l}$ to ensure the natural construction of $B_k$ in our proof can reach similar bound. In other words, we need the separation of covariance matrices to be large as well.
> \
> \
> (__On the SDP theoretical bound__) The theoretical bound is tight in $n$ in the sense that it achieves $\log(n)$, which matches the sample complexity of the original SDP. To address your concern, we have drawn the plots for the relation between separation and mis-clustering rate (Figure 5 in Appendix A.1), where the x-axis is adjusted for different $n$. We can see the same pattern across different settings for $n.$ This shows that the order $\log(n)$ should be tight for the separation bound in Theorem 2.
> \
> \
> (__On the convergence of iLA-SDP__) Currently, we can prove the convergence of iLA-SDP (From Lemma 3 and the construction of oracle LA-SDP we know the objective function is monotonically increasing across the iterations). We also have extensive numerical evidences demonstrating the convergence, which is shown as Figure 6 in the Appendix. However, deriving the convergence rate is quite challenging (as EM algorithm) and it will be our future goal to address it.
> \
> \
> (__On the experiments__) For real datasets, the signal (clsuter separation) is too weak to achieve exact recovery, therefore our theorem does not apply. In fact, we view the theorem as a sanity check for the proposed LA-SDP in the regime of exact recovery. Yes, we want to show the effectiveness of iLA-SDP in the application.

---

### Decision · Program_Chairs · 2023-01-20

**Decision:**

Reject

**Justification For Why Not Higher Score:**

There are no positive votes for acceptance, and though the author response is appreciated, the work would benefit from further review after substantial revisions

**Justification For Why Not Lower Score:**

na

**Metareview: Summary, Strengths And Weaknesses:**

This paper proposes an SDP applied for clustering in a heterogeneous data setting. As reviewers mentioned, there are many such SDP ideas in the literature so the novelty was found to be somewhat limited. There are a number of good ideas behind the approach as mentioned by some reviewers, but unfortunately many issues remained that did not sufficiently convince any reviewers to vote for acceptance. In particular, there is some agreement that the empirical aspects may be the weakest part of this paper, and should be worked on in preparing for a stronger future submission.